# Molecular Simulation of the Binding of Amyloid Beta to Apolipoprotein A-I in High-Density Lipoproteins

**DOI:** 10.3390/ijms26031380

**Published:** 2025-02-06

**Authors:** Chris J. Malajczuk, Ricardo L. Mancera

**Affiliations:** Curtin Medical School and Curtin Medical Research Institute, Curtin University, GPO Box U1987, Perth, WA 6845, Australia; chris.malajczuk@thekids.org.au

**Keywords:** Alzheimer’s disease, amyloid beta, apoA-I, HDL, molecular dynamics simulation

## Abstract

Disrupted clearance of amyloid beta (Aβ) from the brain enhances its aggregation and formation of amyloid plaques in Alzheimer’s disease. The most abundant protein constituent of circulating high-density lipoprotein (HDL) particles, apoA-I, readily crosses the blood–brain barrier from periphery circulation, exhibits low-micromolar binding affinity for soluble, neurotoxic forms of Aβ, and modulates Aβ aggregation and toxicity in vitro. Its highly conserved N-terminal sequence, ^42^LNLKLLD^48^ (‘LN’), has been proposed as a binding region for Aβ. However, high-resolution structural characterisation of the mechanism of HDL–Aβ interaction is very difficult to attain. Molecular dynamics simulations were conducted to investigate for the first time the interaction of Aβ and the ‘LN’ segment of apoA-I. Favourable binding of Aβ by HDLs was found to be driven by hydrophobic and hydrogen-bonding interactions predominantly between the ‘LN’ segment of apoA-I and Aβ. Preferential binding of Aβ may proceed in small, protein-rich HDLs whereby solvent-exposed hydrophobic ‘LN’ segments of apoA-I interact specifically with Aβ, stabilising it on the HDL surface in a possibly non-amyloidogenic conformation, facilitating effective Aβ clearance. These findings rationalise the potentially therapeutic role of HDLs in reducing Aβ aggregation and toxicity, and of peptide mimics of the apoA-I interacting region in blocking Aβ aggregation.

## 1. Introduction

The current prevailing paradigm in Alzheimer’s disease (AD) known broadly as the Amyloid Hypothesis states that all pathological pathways in AD originate from a single, aberrant upstream process involving amyloid beta (Aβ). As such, recent concerted efforts have focussed on elucidating and understanding the influencing factors and species associated with abnormal Aβ during the early, pre-symptomatic stages of disease. To this end, defective lipid/cholesterol metabolism and vascular damage have consistently emerged as critical factors for AD pathogenesis given that their presence significantly enhances late-onset AD (LOAD) susceptibility risk and directly influences Aβ metabolism. This revelation has consequently piqued interest in determining the potential role(s) of lipid- and cholesterol-regulating species such as high-density lipoproteins (HDLs) and their associated apolipoproteins in AD pathology, and more specifically Aβ biochemistry [1].

ApoA-I is the major protein component of circulating periphery HDLs and is the third-most abundant brain apolipoprotein after apoE and apoJ. Yet, unlike apoE and apoJ, apoA-I is not produced locally in the brain but instead enters the central nervous system (CNS) from periphery circulation [2]. Furthermore, there is evidence to suggest that apoA-I transfer from the periphery to CNS may occur in response to abnormal CNS Aβ levels [3]. Recent and accumulating evidence from epidemiological, proteomics, cell culture and animal model studies indicate a neuroprotective role for apoA-I in AD, whereby apoA-I modulates AD risk, severity and the rate of cognitive decline. Moreover, lipidated apoA-I in the form of small, dense lipoproteins are high-affinity substrates for Aβ and act to inhibit Aβ-induced neurotoxicity in a process directly mediated by the apoA-I molecules rather than the lipids bound to it [4,5]. The highly conserved seven-residue ‘LN’ sequence of human apoA-I has been identified as a potential candidate binding site for Aβ owing to its high homology with the high-affinity Aβ-binding ligand ‘GNLLTLD’ (especially for the ‘LN’ residues Asn43-Leu44 and Leu47-Asp48, which are completely conserved across human and a host of analogous mammalian apolipoproteins) [4], and more broadly the known Aβ-transporting abilities of HDLs in the periphery [3,6].

Remarkably, the predominantly hydrophobic ‘LN’ region of apoA-I encompasses a solvent-exposed loop domain between helices in the lipid-free (Δ185–243) apoA-I crystal structure [7]. Whilst this may hold some relevance to the overall proclivity for solvent exposure of the hydrophobic ‘LN’ region, apoA-I in a lipid-bound form as part of a full HDL particle is of far greater biological significance given that apoA-I is rapidly lipidated in circulation. In apoA-I-containing HDLs, the ‘LN’ region represents an overlapping domain between the NT and CT regions of apoA-I that is not structurally or functionally involved in any known HDL maturation processes or reverse cholesterol transport (RCT) pathways beyond a possible role in accommodating changes to particle size [8]. Intriguingly, an earlier study reported that small, protein-rich HDL_3_ particles act as superior substrates for Aβ relative to their larger HDL_2_ counterparts [9], suggesting a subpopulation-specific preferential interaction that could conceivably be modulated by the degree of ‘LN’ exposure at the HDL surface.

Identifying and clarifying the structural and compositional determinants that specifically contribute to Aβ sequestration by apoA-I-containing HDLs would not only provide a compelling insight into the requirements for effective Aβ binding, trafficking, and modulation, but may ultimately aid in the development and refinement of therapeutic interventions targeting AD. However, due to the highly heterogeneous and dynamic nature of both periphery HDLs and Aβ in biology, it is exceedingly challenging to experimentally investigate the structural and dynamic properties of each species in the molecular detail required for rational therapeutic design. By the same token, attempts to identify a specific Aβ-recognition site at the surface of endogenous HDLs or, for that matter, to characterise the underlying biological mechanism(s) of HDL–Aβ interactions in molecular detail experimentally is virtually impossible using available spectroscopic approaches. Consequently, previous attempts to overcome the inherent complexities and thus advance structural and functional understandings for these two species have often centred upon simplified synthetic variants in the form of engineered Aβ mutants and reconstituted HDLs, which have both proven to be valuable (albeit limited) substitutes for each respective biological counterpart given their relative structural and/or compositional homogeneity and thus suitability to experimental spectroscopic analysis.

Engineered Aβ mutants can be designed to exhibit relative structural homogeneity by limiting the overall range of diverse ensembles of discrete conformational substates that wild-type Aβ can otherwise exhibit. In the case of the Aβ_42CC_ mutant, two alanine-to-cysteine substitutions at residues 21 and 30 facilitate the formation of an intramolecular disulphide cross-link, which acts to effectively constrain the peptide in a β-hairpin conformation (i.e., a structural motif involving two contiguous, anti-parallel β-strands that look like a hairpin) that resembles monomeric Aβ bound to the Z_Aβ3_ affibody [10,11]. Moreover, it has been speculated that Aβ exists in a dynamic equilibrium between such a β-hairpin structure and a random coil conformation [12], and that this β-hairpin motif may represent the dominant structural prerequisite for aggregation and neurotoxicity [12,13]. In this respect, the Aβ_42CC_ mutant retains many of the pathologically relevant biophysical and biochemical properties of Aβ_42_, including the ability to rapidly oligomerise into potent, alloform-specific neurotoxic oligomers (without transitioning to higher-order aggregates) [14] that are recognisable to conformation-specific Aβ antibodies [15]. It has been argued that these properties of Aβ_42CC_ provide credibility for this conformational form of the peptide as a viable model for neurotoxic Aβ [14,15,16]. However, in view of the overwhelming conformational diversity exhibited by wild-type Aβ and more broadly in the absence of conclusive evidence for Aβ in AD pathogenesis, it remains an open question as to whether Aβ_42CC_ truly represents a pathologically relevant species of Aβ.

As a potential model for characterising HDL–Aβ interactions, the Aβ_42CC_ mutant is an appealing option given the relatively overwhelming conformational diversity exhibited by wild-type Aβ. A previous study also demonstrated that Aβ_42CC_ mutants preferentially bind to apoA-I derived from cerebrospinal fluid (CSF) and blood serum (and thus presumably in a lipidated form) [17], which is consistent with earlier findings for wild-type Aβ. Additionally, this study reported that many other endogenous proteins that were found to bind to Aβ_42CC_ are also associated with Aβ in senile plaques as well as being encoded by LOAD susceptibility genes, providing a level of confidence that the interactome of Aβ_42CC_ represents a reasonable biological reflection of wild-type Aβ. Taken together, it is conceivable that the conformationally restricted Aβ_42CC_ mutant could act as an appropriate proxy for Aβ within the context of investigating HDL–Aβ interactions.

In a similar way to synthetic engineered Aβ mutants, compositionally homogenous apoA-I-containing discoidal reconstituted HDLs that resemble endogenous nascent HDLs have proven pivotal towards developing detailed structural descriptions for this particular subfraction [18]. However, it has already been established that the synthesis and preparation of homogeneous spheroidal reconstituted HDLs that are compositionally representative of the significantly more abundant and biologically relevant endogenous, mature HDL subpopulations is practically impossible using currently available techniques, for a multitude of factors [19,20]. Hence, computational molecular modelling of HDLs in the form of multiscale molecular dynamics (MD) simulations represents a more viable complementary approach towards predicting, resolving and/or refining the experimentally inaccessible structural and dynamical properties of HDL models at varying levels of detail and complexity [21]. We have recently developed, characterized, and validated representative coarse-grained and atomistic molecular models of the five major periphery HDL subpopulations at two levels of resolution to identify their potential subpopulation-specific structural differences and nuances [22,23,24].

With respect to elucidating the structural determinants for HDL–Aβ binding via these representative HDL subpopulation molecular models, we have argued that the smallest HDL_3c_-sized particles are a relatively more suitable candidate binding partner for Aβ compared with larger apoA-I-containing HDLs due to their smaller particle size, relatively lower lipidation state and consequently higher solvent exposure of ‘LN’ [24]. Specifically for the HDL_3c_ model, its ‘LN’ segments of associated apoA-I chains were frequently identified to be principal constituents of large, contiguous hydrophobic surface patches. Moreover, in the HDL_3c_ model, its ‘LN’ segments typically assumed an extended random coil arrangement whereby solvent exposure of its hydrophobic residues was maximised. Considering the potential for ‘LN’ to specifically interact with Aβ, it is thus conceivable that exposed hydrophobic patches composed primarily of ‘LN’ residues at the surface of small, HDL_3c_ particles could represent a specific binding site for Aβ, wherein hydrophobic interactions drive the mechanism of binding.

We thus hypothesise that the binding of Aβ to HDL particles proceeds through hydrophobic interactions with the ‘LN’ segment of apoA-I. This paper reports a computational molecular simulation study involving extensive MD simulations of an Aβ_42CC_ peptide within the context of the solvent-exposed ‘LN’ segment of apoA-I found in HDL_3c_ particles, representing a feasible and rational approach towards attempting to characterise the possible mechanism of interaction of Aβ with HDL particles. A three-stage MD simulation strategy was devised and implemented to investigate and evaluate the potential for a specific interaction between the ‘LN’ region of apoA-I and an Aβ peptide within the context of a full HDL particle, with the objective of identifying likely interaction modes.

## 2. Results and Discussion

### 2.1. MD Simulation of the ‘LN’ and Aβ42CC Interaction

A 300 ns simulation was conducted to evaluate the time-wise interaction dynamics between the ‘LN’ fragment of apoA-I and the Aβ_42CC_ peptide in aqueous solution. Within 20 ns, ‘LN’ and Aβ_42CC_ had begun surface-level interactions which were maintained for the remaining time (Appendix A). This initial interaction was followed by a progressive average increase in the formation of intermolecular H-bonds for the ensuing 130 ns before stabilising approximately 9–11 H-bonds thereafter (Appendix A). At the same time, transient secondary structures in ‘LN’ were gradually lost following the initial intermolecular interaction, resulting in a random coil arrangement (Appendix A). On the other hand, the secondary structure of Aβ_42CC_ underwent some minor transitions throughout the simulation, predominantly focussed within the unrestrained N-terminal domain preceding the β-hairpin motif and, to a lesser extent, in the loop domain between β-sheets. Notably, secondary structure transitions in Aβ_42CC_ were more prevalent during the first half of the simulation, before a relatively stable peptide secondary structure was established at approximately 150 ns. This coincided with the stabilisation of intermolecular H-bonds, suggesting that, at this time, a stable and persistent complex was formed.

The resulting free energy of interaction landscape for the entire simulation plotted as a function of intermolecular hydrophobic contacts and H-bonds (Figure 1a) reveals the presence of a single minimum corresponding almost exclusively to the second half of the simulation, in agreement with the formation of a stable and persistent complex between ‘LN’ and Aβ_42CC_ following ~150 ns of simulation. With respect to the mode of interaction throughout this period, Figure 1b shows that the Aβ_42CC_ residues surrounding the first β-sheet and including those within the loop region of the induced β-hairpin motif were primarily involved in forming a combination of intermolecular hydrophobic contacts and/or H-bonds with all ‘LN’ residues across the final 150 ns of the simulation. These interacting residues of Aβ_42CC_ also largely corresponded with the non-β-sheet regions, which together assumed a broadly stable secondary structure for the final 150 ns of the simulation. Overall, in this mode of interaction ‘LN’ acts to facilitate the stabilisation of otherwise transiently structured segments of Aβ_42CC_ via the formation of intermolecular hydrophobic contacts and H-bonds with residues surrounding the β-hairpin motif. This prevailing binding mode remains stable until the final trajectory frame, and is illustrated in Figure 1c.

The spontaneous interaction and subsequent formation of an energetically stable complex between ‘LN’ and Aβ_42CC_ during a 300 ns conventional MD simulation run demonstrates that these computational molecular models are fundamentally compatible with each other, consistent with the assertion that apoA-I binding of Aβ_42CC_ mutant peptides can proceed via ‘LN’ [4]. As a prospective binding mode, the resulting ‘LN’–Aβ_42CC_ complex is intriguing on three fronts. Firstly, interaction and subsequent complexation proceeded rapidly, which could be an indication of ‘LN’ exhibiting high-affinity Aβ-binding properties in a similar way to the homologous ‘GNLLTLD’ peptide [4]. Secondly, complex stabilisation proceeded via the formation of a host of intermolecular hydrophobic contacts and/or H-bonds involving all seven ‘LN’ residues. Finally, the formation of this binding mode was dependent on ‘LN’ assuming an extended coil arrangement which resembled the unstructured solvent-exposed loop domain in lipid-free (Δ185–243) apoA-I [7] as well as the structure characterised for ‘LN’ when it represented the primary component of a large contiguous hydrophobic patch at the surface of a HDL_3c_ model [24].

On the other hand, the validity and biological relevance of the resulting mode of interaction between ‘LN’ and Aβ_42CC_ is naturally undermined by the fact that it stemmed from a single, relatively short MD simulation. Specifically, this simulation provided only limited structural sampling for the two species and thus resulted in a relatively underexplored overall conformational free energy landscape beyond the single prevailing mode of interaction. Considering the generally promising aspects of this computational model for ‘LN’-mediated binding of Aβ, it was thus deemed appropriate and necessary to further explore this system using the enhanced sampling T-REMD method.

### 2.2. Identification of Potential Binding Modes for ‘LN’ and Aβ42CC from a T-REMD Simulation

To sufficiently explore the potential modes of interaction between ‘LN’ and Aβ_42CC_, a T-REMD simulation of an equivalent ‘LN’–Aβ_42CC_ system was undertaken across a 48-replica temperature range spanning 298–410 K. All measures for convergence of T-REMD simulations outlined in the Methods section were satisfied across the duration of the simulation. Furthermore, the “time”-wise minimum intermolecular surface distance and combined total surface areas were monitored throughout each replica (a selection of four systems spanning this temperature regime is shown in Appendix A) to specifically measure the convergence of intermolecular interactions and complexation. Accordingly, ‘LN’ and Aβ_42CC_ appeared to exhibit overlapping minimum average values for both quantities after approximately 75 ns of simulation “time” in replicas at 298–310 K, indicating that, within this temperature range, complexation of the two species represented the favourable conformational state for this system hereafter.

Remarkably, ‘LN’ and Aβ_42CC_ also exhibited predominantly interacting conformations for all replicas across the entire temperature range within 150 ns of simulation “time”, as evidenced by the subsequent sustained average minimum intermolecular surface distances for higher temperature replicas (Appendix A). This is somewhat unexpected given that higher temperature replicas correspond with a relative increase in system energy and hence dynamics, which has the potential to destabilise intermolecular interactions such as H-bonds and consequently disrupt complexation. In this respect, increasing temperatures beyond 310 K corresponded with a relatively “slower” rate of association marked by larger fluctuations in the minimum intermolecular surface distances during the pre-association phase of the simulation. Furthermore, during the periods of sustained association the combined average surface area of the interacting peptides progressively increased with increasing temperature (Appendix A). This indicates that the enhanced system dynamics in higher temperature replicas manifested in a general increase in protein solvent-exposure, likely representative of peptide unfolding. However, the progressive increase in peptide unfolding dynamics and solvent exposure did not fully destabilise ‘LN’–Aβ_42CC_ complexes at higher temperatures.

A plausible explanation for the observed affinity between the two species at high temperatures (corresponding to higher free energy states) could be due to the preferential formation of stable intermolecular hydrophobic interactions. Specifically, the strength of the hydrophobic effect has been demonstrated to progressively increase with increasing temperature [25,26,27,28], which has been linked to the IDP phenomenon whereby an increase in temperature can result in a relative contraction of the conformational ensemble [29]. Given the inherent hydrophobic nature of both ‘LN’ and Aβ_42CC_, coupled with an increasing average solvent-accessible surface area with increasing replica temperature, it is conceivable that the induced peptide unfolding at higher temperatures promotes the exposure of hydrophobic regions of each species to interact and form stable intermolecular hydrophobic contacts. Higher free energy conformations in Aβ_42_ have been found to indeed increase exposure of hydrophobic and aromatic residues, consistent with an initial hydrophobically-driven process of aggregation [30].

By contrast, the resulting free energy landscape corresponding to the physiologically relevant replica at 310 K (Figure 2a) exhibits a continuous basin-like form encompassing a wide global minimum which spans 1–13 intermolecular H-bonds and 20–90 hydrophobic contacts, at a relative depth of approximately 8.4 kJ/mol. More broadly, this free energy landscape confirms that the overall conformational sampling of this system is significantly enhanced via a T-REMD regime relative to a conventional MD simulation at the same temperature. Interestingly, the global free energy minimum of the earlier conventional MD simulation exists within the upper limits of the substantially wider global minimum for the corresponding T-REMD replica. Not only does this free energy landscape provide compelling evidence for a considerably wider variety of interaction modes between ‘LN’ and Aβ_42CC_ beyond the prevailing complex identified in the previous conventional MD simulation, but it also highlights the importance of both hydrophobic contacts and H-bonds for the formation and stabilisation of a complex involving these two species.

Figure 2b demonstrates a relative diversity of secondary structures sampled for Aβ_42CC_ and ‘LN’ within the 310 K replica, notwithstanding the general existence of a β-hairpin motif in Aβ_42CC_ encompassing anti-parallel β-strands within the known amyloidogenic ^16^KLVFFC^21^ segment and the C-terminal domain, separated by a loop domain exhibiting high levels of protein curvature (bend structure). Unsurprisingly, the structural variety in this replica is also considerably higher relative to the previous conventional MD simulation to further substantiate the effectiveness of a T-REMD approach towards enhancing the conformational sampling of ‘LN’ and Aβ_42CC_. The relatively high structural diversity observed for ‘LN’ and Aβ_42CC_ in the 310 K replica during the association phase of the simulation (75–400 ns) is consistent with the notion that a wide variety of peptide conformations can be conducive to ‘LN’–Aβ_42CC_ complexation. Moreover, the combination of a broad free energy landscape (Figure 2a) and extensive secondary structure diversity for ‘LN’ and Aβ_42CC_ in the 310 K replica (Figure 2b) suggests that the T-REMD regime was effective in overcoming any potential biases associated with initialising all replicas from the same structural conformations. While future studies may explore initialising replicas from multiple distinct conformations to further enhance sampling efficiency, this observed diversity indicates that this was not a significant limitation in the present study.

A small but potentially significant reduction in helical content within the ‘LN’ fragment was observed after the non-association phase of the simulation (0–75 ns), perhaps indicating the presence of a structural preference for ‘LN’ when associated with Aβ_42CC_. Likewise, the formation of β-strand structure in the C-terminal domain as well as a β-strand spanning the entire ^29^GCIIGLMV^36^ segment of Aβ_42CC_ was only observed during the association phase of the simulation. The latter β-strand structure is particularly interesting in terms of structural determinants for ‘LN’–Aβ_42CC_ binding given that such conformation across this segment of Aβ is consistent with the β-hairpin motif identified as a potentially significant pre-aggregation structural conformation of monomeric Aβ [13] and characterised in the high-resolution NMR structure of Z_Aβ3_-bound Aβ [10,11]. The interaction between ‘LN’ and Aβ_42CC_ thus appears to be associated with small but potentially profound secondary structure preferences for both species. However, the diversity of secondary structures observed for ‘LN’ and Aβ_42CC_ during their interaction could indicate multiple viable binding modes involving a wide range of combinations of intermolecular H-bonds and hydrophobic contacts acting synergistically to stabilise the complex.

The prospect of numerous interaction modes between the ‘LN’ region of apoA-I and Aβ is perhaps not unexpected given the high structural heterogeneity of each species. Furthermore, in the context of preferential HDL-mediated binding and clearance of Aβ it is desirable that this interaction can proceed rapidly and preferentially relative to potentially neurotoxic Aβ self-association pathways. Indeed, Aβ aggregation and neurotoxicity have been demonstrated in vitro to be modulated upon binding to apoA-I at molar ratios of 2:1 Aβ to apoA-I [31]. Thus, a preferential interaction between the two species that is somewhat independent of Aβ conformation (notwithstanding the induced β-hairpin motif in Aβ_42CC_) is consistent with this finding and provides a basis for how this could proceed within a biological context. By the same token, such an interaction between the two species must also act to stabilise Aβ in a non-amyloidogenic state, thus the minor structural tendencies observed for Aβ_42CC_ during the association phase of the simulation could be important towards preventing additional oligomerisation.

Identification and characterisation of potential representative binding modes for ‘LN’ and Aβ_42CC_ could prove valuable towards elucidating the mechanism(s) underlying the neuroprotective interaction of apoA-I-containing HDL particles. To this end, a two-stage clustering regime was employed to explore the extent of conformational heterogeneity in the ensemble of ‘LN’–Aβ_42CC_ complexes within the corresponding free energy basin. Specifically, only the ensemble of conformations corresponding to a free energy less than 8.4 kJ/mol (2 kcal/mol, and representing 83.9% of the simulation) were considered for clustering purposes. As a result, this energy cut-off excluded the non-associating phase of the simulation entirely, focussing exclusively on the low free energy ensemble of conformations sampled throughout the association phase. A schematic showing the overall clustering process and respective secondary structures for ‘LN’ and Aβ_42CC_ across high-occupancy clusters is provided in Figure 3.

The first phase of clustering sought to identify dominant, distinct ensembles of conformations based upon the structure of the ‘LN’ fragment when associated with Aβ_42CC_. Four dominant clusters were obtained totalling 89.3% of the considered free energy surface (74.9% of the entire replica including the non-association phase), with each cluster—herein referred to as C1, C2, C3 and C4—representing 68.4%, 8.8%, 8.2% and 3.9% of the free energy basin, respectively (Figure 3).

Given the reported proclivity for ‘LN’ to exhibit an extended random coil arrangement [7,32,33,34] whilst maximising hydrophobic exposure [24], it is perhaps unsurprising that C1 is composed of conformations where ‘LN’ exhibits a predominantly random coil structure, representing a disproportionate fraction of the considered conformational ensemble relative to the other clusters. Indeed, C2 and C4 are also composed of similarly extended ‘LN’ conformations but with slightly more bend and/or turn structure. The potential significance of this extended conformation of ‘LN’ for Aβ binding was highlighted and discussed above following its identification within the prevailing binding mode from the earlier conventional MD simulation. Consequently, the high representation of this conformation of ‘LN’ throughout the ensemble of low energy states taken from the association phase of the replica further underscores its potential significance for ‘LN’-mediated interactions with Aβ, with clear implications for Aβ binding via small, dense HDLs, wherein a similarly extended ‘LN’ conformation had been frequently characterised in a previous simulation of a representative HDL_3c_ particle [24].

Conversely, C3 is characterised by a prevailing α-helix motif across the ^44^LKL^46^ segment of ‘LN’ and broadly spanning the central five residues (Figure 3), suggesting that an α-helical ‘LN’ structure could potentially also lead to interactions with Aβ_42CC_. A helical tendency for ‘LN’ is unsurprising in view of the dominant α-helical structures observed for lipidated ‘LN’ in HDLs [24], as well as numerous structural [35,36], in vitro [8] and computational [37,38,39] studies that suggest the presence of some helical tendency for the ‘LN’ region. However, within the context of HDL-mediated binding of Aβ, the compatibility of an α-helical ‘LN’ structure for interactions with Aβ remains unclear given the proposition that Aβ binding proceeds preferentially with smaller, protein-rich HDLs because ‘LN’ exposure is greater (and subsequently helical structure is lower) due to fewer stabilising interactions between ‘LN’ and HDL surface lipids. A plausible interpretation of this finding is that for the ensemble of conformations within C3, the α-helical structure of ‘LN’ is stabilised by hydrophobic interactions with the Aβ_42CC_ peptide which mimic those of HDL surface lipids. Therefore, a lipidated α-helical ‘LN’ could nevertheless be incompatible with Aβ given that the requisite residues for interaction are otherwise occupied. Naturally, confirmation of such a model would require additional consideration of the specific binding mode(s) sampled within C3, and perhaps more importantly, the overall proportional relevance of each binding mode with respect to the conformational free energy landscape.

To this end, the second stage of clustering sought to identify five dominant, distinct ensembles of representative ‘LN’–Aβ_42CC_ binding modes based upon the structure of Aβ_42CC_ within the four highest-occupancy ‘LN’ clusters C1, C2, C3 and C4. Interestingly, the Aβ_42CC_ secondary structure profiles corresponding to the highest-ranked subclusters provide further evidence for the notion that ‘LN’ can bind to Aβ_42CC_ in a manner that is somewhat independent of Aβ_42CC_ conformation. This is best demonstrated for the highest-occupancy subcluster of C2 (C2-1) where Aβ_42CC_ exhibits significant helical structure throughout its C-terminal domain as well as the central ^23^DVGSNKG^29^ region that would otherwise represent the loop domain within the disulphide-induced β-hairpin motif. Indeed, the β-hairpin motif is practically absent within the ensemble of conformations corresponding to C2-1. However, the significance of such a binding mode is somewhat contextualised by the fact that it only represents 1.6% of the total considered conformational ensemble. As such, subclusters were ranked based upon their overall proportional representation relative to the total considered conformational ensemble and the five highest binding modes for ‘LN’–Aβ_42CC_ were selected for further interrogation.

As shown in Figure 3, four of the five representative binding modes stemmed from the C1 ensemble, representing 8.7% (C1-1), 6.0% (C1-2), 5.1% (C1-3) and 2.9% (C1-4) of the total considered conformational ensemble. Conversely, the fifth-highest-occupancy binding pose was identified as the top-ranked cluster within C3 (C3-1), to represent 2.7% of the total considered conformational ensemble. It is unsurprising that the top four binding poses originated from within C1 given that this ensemble of conformations represented more than two-thirds of the considered free energy landscape. On the other hand, the C3-1 binding mode is characterised by a prevailing α-helical structure for ‘LN’, indicating that this prospective binding pose represents a small but significant proportion of all low free energy interactions. Given the high structural heterogeneity of both species, and mounting evidence that ‘LN’ can interact with Aβ_42CC_ in a variety of ways, it is unsurprising that the proportional relevance of each representative binding mode is relatively small. However, as a representative subset of prospective ‘LN’–Aβ_42CC_ binding poses, these five conformations account for more than a quarter of the total considered conformational free energy landscape (25.4%). Moreover, as demonstrated in Figure 4, each binding pose reflects a distinct average interaction registry between ‘LN’ and Aβ_42CC_ involving numerous intermolecular hydrophobic contacts and H-bonds across a range of residues and including a host of different secondary structure profiles, providing a basis for evaluating their potential as possible binding modes within the context of ‘LN’-mediated binding of Aβ_42CC_ to a full HDL_3c_ model.

### 2.3. Evaluation of ‘LN’–Aβ42CC Binding Poses Within the Context of a Full HDL Particle

It is important to assess the thermodynamic stability and general suitability of representative ‘LN’–Aβ_42CC_ binding poses within the context of an HDL-mediated interaction with Aβ, and more broadly to evaluate the structural determinants that contribute to favourable binding between the species. The five highest-occupancy binding poses identified above, as well as the final bound structure from the previous conventional MD simulation (herein referred to as the cMD binding pose), were each grafted to individual HDL_3c_ model particles and simulated for 500 ns. In all cases, the Aβ_42CC_ peptide remained complexed with the HDL_3c_ model across each simulation, demonstrating that the six starting modes of interaction are compatible with forming a stable HDL_3c_-Aβ_42CC_ complex (Figure 5). Moreover, the final configuration of each simulation showed Aβ_42CC_ adorning the surface of the HDL_3c_ particle in proximity to its opposing ‘LN’ region as well as other apoA-I chains on the particle surface, as illustrated in Figure 5. An exterior protein–protein interaction between HDLs and Aβ is consistent with the general premise that HDLs carry and transport a variety of protein “decorations” at their surfaces [40], as opposed to a partial embedment of the Aβ peptide into the lipid phase of the particle, which has been proposed to occur in larger lipoproteins [41]. To some extent, this observed mode of Aβ-sequestration via small HDLs can be rationalised by the earlier findings that the surface of HDL_3c_ is overwhelmingly composed of protein [22,24] and that it exhibits a high relative surface tension [23], which may act to inhibit Aβ insertion.

**Figure 4 ijms-26-01380-f004:**
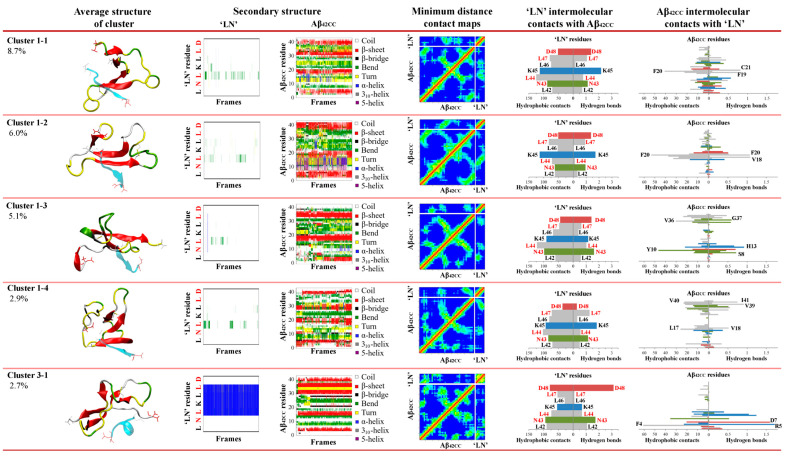
Secondary structures and the average intermolecular properties for the top five binding poses between the ‘LN’ fragment of apoA-I and an Aβ_42CC_ peptide from a T-REMD replica system at 310K. Each binding pose exhibits unique intermolecular interactions between ‘LN’ and Aβ_42CC_ as highlighted by the distinctive minimum distance contact maps and Aβ_42CC_–‘LN’ intermolecular interaction residues. Note: Contact map colouring reflects a red to royal blue rainbow mapping scheme where prevailing contacts are represented by red and no contact is represented by blue, with residue numbering beginning from the bottom-left of the axis; horizontal bar plots are coloured according to general residue typing, with select residues labelled according to their single-letter amino acid code followed by their residue number.; red lettering in ‘LN’ secondary structure plots and ‘LN’ intermolecular contact bar plots correspond to the residues within ‘LN’ region of apoA-I that are equivalent to the high-affinity Aβ-binding ‘GNLLTLD’ peptide.

The favourability of each binding pose was evaluated by computation of the free energies of binding across each HDL–Aβ_42CC_ simulation using the MM-PBSA approach. The resulting binding free energies, averaged over 100 ns intervals, are presented in Figure 6a and demonstrate that all binding modes exhibited some instability throughout each simulation. Of the six initial binding poses, only the C1-1 and C1-4 poses appeared to stabilise to a steady value within 500 ns. In both cases, these modes of interaction remained favourable across the simulation duration. On the other hand, C1-3 and C3-1 exhibited unfavourable binding free energies throughout their respective simulations. The remaining two representative binding modes exhibited periods of positive and negative average binding free energy values across the 500 ns of simulation, with the initially unfavourable C1-2 pose ultimately having a negative binding free energy in the concluding 100 ns interval, whilst the cMD pose became unfavourable within this final interval.

The overall variability and instability of the free energy of binding estimates across the six simulations is expected for a multitude of factors, not the least of which includes the unique initial interaction registry for each binding pose and the fact that each pose was rationalised in the absence of a proximal HDL surface, which could be reasonably expected to influence the ‘LN’–Aβ_42CC_ interaction. In addition, a 500 ns conventional atomistic MD simulation is likely inadequate to fully capture the highly heterogeneous structure of Aβ_42CC_ (as evidenced by the earlier findings for a conventional MD simulation of ‘LN’ and Aβ_42CC_) or account for the slow surface dynamics inherent to an HDL_3c_ particle [22,23], let alone be expected to comprehensively characterise HDL–Aβ_42CC_ interactions. Hence, it is important to emphasise that the overall purpose of these simulations and the subsequent MM-PBSA analysis was to identify and explore potential structural determinants that underpin favourable and unfavourable binding between the two species across six distinct representative binding modes, as opposed to attempting to obtain definitive modes of interaction between HDL and Aβ_42CC_.

To assess the factors that contributed to the favourability of each binding mode, decomposition of the average free energies of binding for the major system components was undertaken across the final 100 ns of each simulation and is plotted in Figure 6b. In general, the absolute contributions via Aβ_42CC_ and/or ‘LN’ represented the most influential components of the overall free energies of binding. On the other hand, HDL lipids conferred a largely insignificant influence on the overall free energies of binding for each binding mode. These outcomes are expected given that each binding pose had been rationalised upon the basis that HDL-mediated interactions with Aβ are initiated specifically via an exposed ‘LN’ region. Furthermore, the low contribution of HDL lipids to the overall free energies of binding is not surprising considering the relatively small surface area that HDL lipids occupy across the HDL_3c_ interactome [22,24].

Whilst Aβ_42CC_ and/or ‘LN’ represented the principal influencing components to the overall free energies of binding in five of the six simulations, for the ultimately unfavourable cMD mode of interaction, a high positive free energy of binding via the succeeding residues of the apoA-I chain containing the interacting ‘LN’ segment counteracted a considerable negative free energy contribution from Aβ_42CC_. This unfavourable free energy contribution was traced to three charged residues within the apoA-I C-terminus (E223, E234 and K238, see Appendix A) which reside in the general vicinity of the ‘LN’ region and are each implicated in the formation of stabilising salt-bridges with opposing anti-parallel apoA-I chains according to the trefoil arrangement for apoA-I [42]. Surprisingly, during the MD simulation these three charged C-terminal residues underwent preferential dissociation from their corresponding salt-bridge trefoil registry to form transient intermolecular H-bonds with Aβ_42CC_. Yet the relative instability of these subsequent interactions during the final 100 ns, together with the partial decoupling of this section of the trefoil arrangement, resulted in a net increase in binding free energy for this pose during the concluding 100 ns interval. Considering the inherent limitations of these simulations outlined above, it would be reticent to speculate that this observed behaviour for Aβ_42CC_ represents a real destabilising mechanism of action to the trefoil arrangement, particularly given that (1) the free energy of binding values fluctuated considerably throughout this simulation, suggesting that the binding mode had not stabilised by the final 100 ns interval, and (2) a similar preferential decoupling of the trefoil arrangement was not observed in the other simulations with different initial binding poses. However, the underlying premise that HDL–Aβ binding proceeds via a multi-step mechanism initiated via a specific ‘LN’–Aβ interaction and subsequently stabilised by other charged apoA-I residues on the HDL surface is conceivable.

Given the importance of ‘LN’ and Aβ_42CC_ to the overall free energies of binding estimated for all binding modes, residue-wise decomposition across both molecules was undertaken to elucidate which residues contributed significantly towards binding. Figure 7 shows that the contributions from Asp48 of ‘LN’ and Lys16 of Aβ_42CC_ appear to confer the most disparate influence on the overall favourability of the free energies of binding for each binding mode. Specifically, relatively low to negligible contributions from Asp48, and low or negative contributions from Lys16 were associated with the three favourable binding modes (Figure 7a). On the other hand, the cMD interaction mode exhibited the highest contribution from Asp48 in ‘LN’ (Figure 7b), whilst large positive energy contributions in both residues were associated with an overall unfavourable free energy of binding for C1-3 and C3-1. To a lesser degree, relative negative energy contributions from non-polar residues across the C-terminus of Aβ_42CC_ (Val36, Val40 and Ile41) also corresponded with the three favourable binding modes.

In view of the importance of Asp48 and Lys16 to binding, it is intriguing to consider that both residues have been implicated in modulating toxic Aβ aggregation pathways. With respect to Asp48 of ‘LN’, the importance of this residue to ‘LN’-mediated Aβ binding and modulation was previously highlighted due to its strict conservation across human and mammalian apolipoprotein sequences [38] coupled with its presence within the homologous ‘GNLLTLD’ peptide (which acts as a high-affinity binding ligand for Aβ) [4]. In the case of Lys16, this residue in Aβ represents the first (and only charged) amino acid within the amyloidogenic ^16^KLVFF^20^ segment, which is strongly implicated in toxic Aβ aggregation [43]. Moreover, the ^16^KLVFF^20^ segment of Aβ has served as a template for the rational design of compounds that disrupt toxic Aβ aggregation [43], formulated under the assumption that the spontaneous self-association of Aβ occurs in a process that seeds the formation of toxic higher-order aggregates containing a β-hairpin structural conformation [44,45]. It is thus interesting to consider that in Aβ_42CC_, the ^16^KLVFF^20^ segment comprises most of the first β-strand within the β-hairpin motif. More curious is the fact that the three clusters that exhibited a high relative proportion of intermolecular interactions between ‘LN’ and Aβ_42CC_ residues located within the ^16^KLVFF^20^ segment (C1-1, C1-2, and C1-4; Figure 4) correspond with the three ultimately favourable binding modes following simulation as part of an HDL_3c_-Aβ_42CC_ complex.

To further interrogate the three favourable binding modes and ultimately explore the influence of Asp48 of ‘LN’ and the ^16^KLVFF^20^ segment with respect to binding, Figure 8 shows a snapshot of the final configuration of the binding mode and the secondary structure profiles for the interacting ‘LN’ segment and Aβ_42CC_ for HDL simulations containing C1-1, C1-2 and C1-4. An interesting observation is that the position of the Aβ_42CC_ peptide with respect to ‘LN’ is dissimilar across the three binding modes, with ‘LN’ positioned atop, beneath and to the side of Aβ_42CC_ in C1-1, C1-2 and C1-4, respectively. Given that all three poses share a similar favourable free energy of binding value during the final 100 ns of simulation (Figure 6a), this could suggest that for these poses the orientation of ‘LN’ does not significantly influence the overall stability of Aβ_42CC_ binding. Furthermore, the broadly distinct (albeit largely unchanging) secondary structure profiles for Aβ_42CC_ across the three simulations shown in Figure 8 provides further evidence that ‘LN’ can bind to Aβ_42CC_ in a variety of ways and in a manner that is somewhat independent of Aβ_42CC_ conformation. The caveat is that at least some β-strand conformation exists within the ^16^KLVFF^20^ segment of Aβ_42CC_ in each of these interaction modes. By contrast, the virtual absence of secondary structure for ‘LN’ in each pose represents a clear structural determinant for favourable binding in these simulations.

A per-residue analysis of the final binding interfaces between ‘LN’ and Aβ_42CC_ corresponding to the three favourable binding modes in Figure 9 reveals that a diverse combination of intermolecular hydrophobic and H-bonded interactions between the two species, as well as hydrophobic HDL surface lipid interactions with ‘LN’, act together to stabilise each binding interface. Interestingly, hydrophobic interactions between HDL surface lipids and the potentially critical Asp48 residue of the interacting ‘LN’ segment represent the only commonality for this residue across the three favourable binding modes, consistent with Asp48 being in the closest relative proximity to the HDL surface when ‘LN’ is in an exposed extended arrangement [24]. Whilst this finding does not imply a direct role for Asp48 in the binding of Aβ, as had been suggested in a previous study [4], it could be suggestive that Asp48 undertakes a structural role by anchoring the apoA-I chain at the HDL surface whilst ‘LN’ is exposed for binding.

An “anchoring” mechanism for Asp48 of the interacting ‘LN’ segment of apoA-I is consistent with the recent finding that Asp residues form strong preferential interactions with PC headgroups (the major surface lipid species in HDLs) and especially with anionic lipids [46], which incidentally are enriched in the small, dense HDL_3c_ subfraction [47]. At the same time, such an arrangement for Asp48 is not consistent with the relative position of negatively charged sidechains in a Class-A AαH motif (directly opposite the non-polar face of the helix oriented towards the solvent), which has previously been assigned to the encompassing 22-residue H1 domain of lipidated apoA-I [48]. However, it is interesting to consider that an unravelling of the helix within the preceding ‘LN’ segment—such as was characterised for this region when solvent-exposed—considerably alters the position of Asp48 with respect to the prevailing non-polar face within this AαH, bringing it to a position that is relatively adjacent to the HDL interface and ostensibly in a position to form interactions with HDL surface lipid headgroups (Figure 10). Incidentally, this positional difference for Asp48 may have also contributed to the unfavourable binding of the C3-1 binding pose, wherein ‘LN’ retains some helical structure. Taken together, it is conceivable that preferential interactions between Asp48 and HDL surface lipids act to facilitate ‘LN’ exposure whilst stabilising the succeeding residues within the C-terminal domain of apoA-I on the HDL surface, ensuring that the overall particle structure and stability is retained following Aβ binding.

Concerning the ^16^KLVFF^20^ segment of Aβ_42CC_, Figure 9 reveals that multiple residues located within this region were centrally involved in forming intermolecular H-bonds and hydrophobic interactions with ‘LN’ for C1-1 and C1-2. However, commonality between interacting residue pairs was not observed across the two binding modes in a finding broadly consistent with the notion of a non-specific binding mechanism for ‘LN’ and Aβ involving the ^16^KLVFF^20^ segment. Considering these findings for simulations of HDL–Aβ_42CC_ complexes containing the C1-1 and C1-2 poses, in combination with the fact that the original C1-1 and C1-2 subclusters represented the two highest-occupancy subclusters (together accounting for almost 15% of the considered free energy surface from the T-REMD simulation described above), there is reason to suspect that the ^16^KLVFF^20^ segment in a β-structure could represent a specific binding site for the ‘LN’ segment of apoA-I. In this manner, it is also conceivable that a preferential interaction between the ^16^KLVFF^20^ segment via ‘LN’ not only represents a stable and favourable binding mode but could act to preclude further oligomerisation of Aβ by virtue of the interacting ‘LN’ blocking this potential nucleation site of Aβ [31]. Such an action could begin to rationalise earlier findings showing that lipid-free apoA-I and apoA-I-containing HDL effectively modulate Aβ aggregation [4].

Alternatively, the relative absence of interactions involving the ^16^KLVFF^20^ segment of Aβ_42CC_ in the favourable C1-4 binding mode may also highlight a multifaceted quality to ‘LN’-mediated binding of Aβ. Figure 9c shows that only Val18 of the ^16^KLVFF^20^ segment of Aβ_42CC_ contributed hydrophobic contacts to the ‘LN’ binding interface in the binding mode for the HDL–Aβ_42CC_ complex containing the C1-4 pose. Closer inspection of this binding mode at the end of this simulation in Figure 8c shows Aβ_42CC_ in a relatively solvent-exposed conformation compared with the other two favourable interaction modes, yet the ^16^KLVFF^20^ segment of Aβ_42CC_ appears to be oriented towards the HDL surface. Assuming that this segment of Aβ is indeed a nucleation site for the formation of neurotoxic aggregates, it may be speculated that the C1-4 interaction mode indirectly renders Aβ_42CC_ incompetent for further oligomerisation by aligning the ^16^KLVFF^20^ segment away from the solvent. At the same time, subsequent interactions between the ^16^KLVFF^20^ segment and other interaction sites on the HDL surface following ‘LN’-mediated binding may also act to stabilise Aβ_42CC_ in a non-amyloidogenic state broadly consistent with a multi-step mechanism for binding between HDL and Aβ.

## 3. Materials and Methods

### 3.1. The Aβ_42CC_ Model and the ‘LN’ Fragment

The monomeric Aβ_42CC_ model was modelled based on the structure of monomeric Aβ_40_ bound by the Z_Aβ3_ affibody (Protein Databank ID 2OTK), with a β-hairpin conformation in the Lys16-Phe20 and corresponding Ile32-Val36 regions. Alanine-to-cysteine substitutions at residues 21 and 30, as well as peptide extension to include the hydrophobic Ile41 and Ala42 C-terminal residues, were performed using BIOVIA Discovery Studio v. 4.0 (Dassault Systèmes, Paris, France) software and output to a PDB structure file for use with the GROMACS v. 4.6.7 MD simulation package [49]. The addition of a disulphide bridge between Cys21 and Cys30 was specified within GROMACS v. 4.6.7. Weak distance restraints were then applied to key atoms within each Aβ_1–42_CC monomer consistent with ssNMR data to ensure that the intramolecular interactions corresponding to the β-hairpin conformation of Aβ_1–42_CC were initially established in the starting conformation. Distance restraints were then gradually removed from the monomer over 2.0 ns during equilibration. The β-hairpin conformation remained stable across the remaining simulation time after all distance restraints were removed. Titratable protein residue groups were assigned their typical protonation states at pH 7. Specifically, histidine residues His6, His13 and His14 were each protonated on the δ-nitrogen group consistent with this isomer of Aβ being strongly associated with the β-hairpin arrangement [50].

The ‘LN’ fragment of apoA-I was modelled as an extended coil using BIOVIA Discovery Studio v. 4.0 (Dassault Systèmes, Paris, France) software and output to a PDB structure file for use with the GROMACS v. 4.6.7 MD simulation package [49]. Termini residues were assigned a neutral charge (NH_2_ and COOH for N- and C-termini, respectively) to better reflect the natural state of ‘LN’ as a segment confined within the apoA-I protein.

### 3.2. HDL Particles

We have recently rationalised that the HDL_3c_ subfraction is the most likely candidate for ‘LN’-mediated Aβ binding [24]. A candidate structure of the HDL_3c_ model which exhibited the maximum hydrophobic exposure of the ‘LN’ region of an apoA-I chain (see Figure 9 in [24]) was isolated and extracted from the previously reported simulation trajectory of HDL_3c_ in water and used for grafting of Aβ_42CC_-LN binding poses. Despite being a relatively low-abundance conformation of the ‘LN’ region within the context of a full HDL model, this structure of HDL_3c_ was deemed to be most amenable to grafting given the high relative exposure of ‘LN’ and hence lower propensity for steric clashes between Aβ_42CC_ and the HDL surface upon grafting. Full details of the construction and coarse-grained and atomistic simulation of the five-lipid HDL subpopulation models have been previously reported [22,24]. The five-lipid models of these HDL subpopulations contain multifoil apoA-I arrangements and include 1-palmitoyl-2-oleoyl-*sn*-glycero-3-phosphocholine (POPC), lysophosphatidicylcholine (LysoPC), trioleate (TO), cholesterol (CHOL) and cholesteryl oleate (CO). They were shown to exhibit size, morphological, spatial, and compositional profiles that are consistent with experimental quantities corresponding to their respective subpopulation.

### 3.3. Simulation Details and Analyses of Data

Atomistic conventional MD simulations and T-REMD simulations were performed with the Gromacs 4.6.7 package [49]. The standard GROMOS 54A7 united-atom parameter set [51] was used to describe protein molecules and counter-ions (Na^+^ and Cl^−^). POPC, LysoPC and CHOL were modelled using the GROMOS 53A6L force field (which forms part of the GROMOS 54A7 parameter set) [51,52]. United-atom models of CO and TO consistent with a modified version of the GROMOS 54A7 force field were generated by the Automated Topology Builder v. 2.2 [53]. Water molecules were described by the SPC model [54].

The GROMOS53A6 force field has been previously demonstrated as an appropriate choice for reproducing the given β-hairpin propensity of an Aβ_21–30_ fragment by suppressing atypical helical structure [55]. The chosen force field for our study, GROMOS54A7, is a re-parameterisation of the GROMOS53A6 that includes a slight re-adjustment of the torsional angle terms for better reproduction of protein helical propensities. Given this re-parameterisation of GROMOS53A6, the performance of GROMOS54A7 to sample and stabilise the natural folding of β-peptides into β-hairpin conformations has been thoroughly tested and validated as a suitable force field for simulations of β-peptides containing less than two helical folds [56,57,58]. It should be acknowledged that the GROMOS54A7 force field exhibits some conformational bias towards a stable bound state-like structure in very long simulations (~63 µs) of systems containing multiple intrinsically disordered proteins [59]. Hence, it was deemed crucial that T-REMD simulations were also undertaken in this work to limit this effect and enhance the level of conformational sampling of the Aβ_42CC_ peptide.

A single conventional MD simulation spanning 300 ns was initially conducted on a system composed of one Aβ_42CC_ monomer and one ‘LN’ fragment of apoA-I spaced 2.0 nm apart in aqueous solution and at an ionic strength of 0.1 M, to explore the interaction dynamics between the two species and evaluate the extent of conformational sampling. This was followed by temperature replica exchanged molecular dynamics (T-REMD) simulations to sufficiently explore the conformational landscape and potential binding modes for this system. For this a series of 48 replica systems containing one Aβ_42CC_ monomer and one ‘LN’ fragment of apoA-I in water were simulated across a temperature range spanning 298–410 K for 400 ns each, totalling 19.2 µs of simulation. Replica temperatures were calculated according to the algorithm of Patriksson and van der Spoel [60] to give an estimated exchange probability of 20%. A temperature scheme that included replicates at temperatures below the reference temperature of 310 K was used to account for the counterintuitive IDP phenomenon whereby a reduction in temperature can result in a relative expansion in the conformational ensemble [29]. Intermittent pair-wise exchanges of system states between simulations at adjacent temperatures was undertaken according to the Metropolis criterion when the potential energy distributions for the two systems overlap sufficiently to satisfy a given acceptance ratio. Exchange attempts were made every 500 steps.

General convergence of the T-REMD simulation was determined according to the broad criteria that (1) a high and uniform conformational exchange ratio between adjacent replicas was maintained throughout the simulation, (2) each replica traversed the entire temperature range on multiple occasions, and (3) replicas did not become trapped in local minima [61]. Accordingly, each condition was satisfied across the duration of the presented T-REMD simulation. Specifically, the final average exchange probability for the simulation was calculated as 22 ± 1%.

Following a two-step clustering regime (as described further below), five representative poses of Aβ_42CC_ bound to the ‘LN’ fragment were identified and isolated from the 310 K replicate in the T-REMD simulation. A sixth Aβ_42CC_-‘LN’ binding pose was isolated from the final frame of the initial conventional MD simulation run for comparison.

Each selected representative pose for Aβ_42CC_ bound to the ‘LN’ fragment was grafted to individual HDL_3c_ model particles following alignment of protein mainchain atoms from the ‘LN’ fragment with its corresponding atoms found in the exposed ‘LN’ region of the HDL_3c_ structure. Goodness of fit was assessed visually via VMD v. 1.9.3 ensuring that steric clashes were minimised. Grafted HDL–Aβ_42CC_ complexes were then each centred in a dodecahedron box and hydrated with 45,000–55,000 water molecules, ensuring a minimum distance from the surface of the complex to the simulation box edge of 2.5 nm. Counter-ions were added to neutralise the charge of each HDL–Aβ_42CC_ system and an ionic strength of 0.1M was included in each system to ensure consistency with experimental conditions [4]. Steepest descents energy minimisation was undertaken on each HDL–Aβ_42CC_ complex system, followed by short (100 ps) NVT and then NPT simulations, each conducted at 210 K. A subsequent NPT equilibration (2.0 ns) at 310 K was conducted prior to 1.0 µs of production for each binding pose.

Lipids, protein and solvent (water + counter-ions) were weakly coupled separately to a temperature bath at 310 K using the Nosé–Hoover extended ensemble thermostat [62,63] with a coupling time constant of 0.1 ps. Pressure was maintained at 1 bar isotropically with the Parrinello-Rahman barostat [64] with a time constant of 2.0 ps. Electrostatics were handled by the particle mesh Ewald (PME) scheme [65] with a 0.8 nm real-space cut-off, a 0.12 nm reciprocal space gridding and splines of order 4. The neighbour grid searching method was applied and the neighbour list was updated each time step. The LINCS algorithm was used to constrain bond lengths [66]. The equations of motion were integrated using a leap-frog algorithm with a time-step of 2.0 fs.

A variety of the Gromacs 5.0.2 tools were used to evaluate the equilibrium properties of each system across the simulation protocol. To evaluate complexation during simulations of the ‘LN’ fragment of apoA-I and Aβ_42CC_, a variety of intra- and intermolecular properties were assessed as a function of simulation time. Specifically, the minimum surface-to-surface distance between the two species was computed using the ‘*gmx mindist*’ program, whereas centre-of-mass distances were calculated using the ‘*gmx distance*’ program. The number of hydrogen bonds between the ‘LN’ fragment of apoA-I and Aβ_42CC_ were calculated using the following definition of a hydrogen bond (H-bond): the maximum distance between a H-bond donor and its acceptor was defined to be *r* = 3.25 Å, and the maximum angle between the donor–acceptor vector and the donor–hydrogen atom vector was defined to be ∠35° [67]. The dynamic secondary structure of the ‘LN’ fragment of apoA-I and Aβ_42CC_ was calculated using DSSP via the ‘*gmx dssp*’ wrapper. Hydrophobic intermolecular contacts were defined as non-polar atom pairs within a cut-off of 4.5 nm. The ‘LN’–Aβ_42CC_ total complex SASA was calculated with ‘*gmx sasa*’ using a probe radius of 0.14 nm.

With respect to the presentation of time-wise results calculated from individual replicas within the T-REMD simulation, it is important to emphasise that simulation time is not reflective of a continuous unbroken trajectory due to the intermittent conformational exchanges between adjacent replicas. Hence, for convenience, the progression of the T-REMD simulation is expressed as “time” throughout the relevant discussion section as an acknowledgment of the ambiguity in this quantity.

To isolate potential binding modes between ‘LN’ and Aβ_42CC_, a two-step clustering regime was followed for all low-energy conformations isolated from the T-REMD simulation trajectory at 310 K (ΔG < 2 kcal/mol, representing 83.9% of the total simulation trajectory). Clustering was undertaken via the *gmx cluster* program using the ‘Gromos’ algorithm [68]. The first stage of clustering was centred upon the ‘LN’ fragment using a C_α_ RMSD cut-off of 0.2 nm to yield four major representative conformations for ‘LN’. A subsequent clustering of these four major ‘LN’ clusters ensued, centering upon the Aβ_42CC_ peptide using a C_α_ RMSD cut-off of 0.6 nm. C_α_ RMSD cut-offs at both stages of the clustering regime were chosen following a stepwise variation in C_α_ RMSD cut-offs between 0.1 and 1.0 nm in steps of 0.05 nm, with final values decided according to the overall quantity and variety of dominant clusters obtained and ensuring that the resulting dominant clusters represented a reasonable proportion of the total ensemble. The resulting clustering subsets were then ranked according to highest occupancy as a percentage of the considered conformational space, with the average structure for the five highest ranked cluster subsets isolated for further evaluation as potential Aβ_42CC_-‘LN’ binding poses. These structures were then grafted to individual HDL_3c_ particles and each simulated for 500 ns to evaluate their suitability as binding poses.

The free energies of binding for the five poses plus the final structure from the conventional MD simulation were approximated across HDL–Aβ_42CC_ simulation trajectories using a single-trajectory MM-PBSA approach via the open-source *g_mmpbsa* v. 1.6 tool [69]. The choice to use a single-trajectory approach for predicting the free energies of binding was based upon the relative conformational stability of the HDL_3c_ particle and the Aβ_42CC_ peptide throughout the simulation time. The free energy of binding was calculated intermittently across each 500 ns simulation at an interval of 1.0 ns, totalling 500 conformations per binding mode. Average residue-wise contributions to the total binding energy were estimated via the MmPbSaDecomp.py Python v. 2.7 script that comes packaged along with the *g_mmpbsa* v. 1.6 tool. We note that the MM-PBSA approach does not include a computation of the change in entropy (i.e., vibrational, translational and rotational), which would require additional calculations. It is important to acknowledge that enhanced sampling methods (such as umbrella sampling) are available for the prediction of the association and/or dissociation of molecular complexes; however, the computational effort of such approach would be prohibitive given the large size of the systems investigated here.

## 4. Conclusions

This study aimed to investigate and evaluate ‘LN’-mediated interactions with the Aβ_42CC_ mutant peptide following a three-step MD simulation procedure to identify potential structural determinants for HDL–Aβ binding. Firstly, a conventional MD simulation of the isolated ‘LN’ fragment and the Aβ_42CC_ peptide in aqueous solution confirmed the occurrence of a spontaneous and stable interaction between the two molecules whilst highlighting the sampling limitations inherent in this approach. Subsequently, a T-REMD simulation approach was utilised to enhance the overall conformational sampling between the two interacting molecules within an equivalent solvated ‘LN’–Aβ_42CC_ system. The five highest-occupancy binding modes between ‘LN’ and Aβ_42CC_ were isolated from the free energy basin corresponding to a T-REMD replica system at physiological temperature (310 K) according to a two-step clustering regime. A sixth potential binding mode between ‘LN’ and Aβ_42CC_ was extracted from the final configuration of the earlier conventional MD simulation for comparative purposes. In five of the six binding modes, the ‘LN’ region exhibited a predominantly random coil conformation whilst the remaining pose included an α-helical ‘LN’ fragment. With respect to the secondary structures of Aβ_42CC_ across the six binding poses, the imposed β-strand tendencies within the central hydrophobic core and the C-terminus represented the only consistent secondary structure feature across the six ‘LN’–Aβ_42CC_ binding poses.

Each binding pose was then individually grafted in place of the exposed ‘LN’ section of an apoA-I chain of an atomistic HDL_3c_ model. The six subsequent HDL–Aβ_42CC_ complex systems were then subjected to conventional MD simulations to assess the stability of each binding mode within the context of a full HDL_3c_ particle. In all cases, Aβ_42CC_ remained complexed with HDL_3c_ via ‘LN’ for the duration of each MD simulation. By the end of the simulations, each bound Aβ_42CC_ peptide appeared to have collapsed onto the HDL surface in contact with the interacting ‘LN’ region as well as other apoA-I chains. Calculation of the free energies of binding for each Aβ_42CC_-HDL_3c_ complex revealed that three of six binding modes ultimately exhibited a favourable interaction. In all binding modes, the interacting ‘LN’ and Aβ_42CC_ species conferred significant influence on the overall free energy of binding between Aβ_42CC_ and HDL_3c_. Per-residue decomposition of the free energy of binding for ‘LN’ and Aβ_42CC_ revealed that Asp48 of ‘LN’ and Lys16 within the amyloidogenic ^16^KLVFF^20^ segment of Aβ_42CC_ had a disparate contribution to the overall free energies of binding across the six binding modes.

For the three favourable binding modes, ‘LN’ exhibited an almost exclusively extended random coil conformation, whereas the secondary structure of Aβ_42CC_ was unique to each binding mode apart from some β-strand conformation within the central ^16^KLVFF^20^ region. Moreover, intermolecular hydrophobic and H-bonded interactions between the interacting ‘LN’ and Aβ_42CC_ residues and often involving the ^16^KLVFF^20^ segment acted to stabilise each binding mode, with minor contributions from the HDL lipid surface occurring in all cases and specifically with the Asp48 residue of ‘LN’. Intriguingly, the loss of helical conformation in the preceding ‘LN’ segment of apoA-I modified the position of Asp48 relative to the non-polar face of the subsequent AαH to enhance the availability of Asp48 to form interactions with HDL surface lipid headgroups and potentially stabilise the apoA-I arrangement on the HDL surface during ‘LN’–Aβ binding.

Overall, favourable binding of Aβ by HDLs was found to be driven by a combination of intermolecular hydrophobic and H-bonded interactions predominantly between the ‘LN’ segment of apoA-I and Aβ. Specifically, four major structural determinants were identified to significantly contribute to favourable binding: (1) ‘LN’ exposure from the HDL surface in an extended coil arrangement, (2) β-strand conformation across the ^16^KLVFF^20^ region of Aβ, (3) lipid interactions with the Asp48 residue of ‘LN’, and (4) HDL interactions with the ^16^KLVFF^20^ segment, either directly via ‘LN’ or through contacts stabilising this segment at the HDL surface. It is interesting to note that the ^16^KLVFF^20^ segment of Aβ has long been regarded as a critical seeding domain for amyloidogenesis [70,71] and to this day remains an active target for therapeutics designed to control Aβ toxicity [72,73,74]. Hence, preferential interactions between HDLs and Aβ that limit the solvent-exposure of ^16^KLVFF^20^ may thus stabilise Aβ in a non-amyloidogenic conformation on the HDL surface, reducing its ability to oligomerise and form neurotoxic aggregates, and allowing for effective Aβ clearance. However, the role of the amyloidogenic ^16^KLVFF^20^ region in the initial stages of aggregation is unclear, so whether the conformation of Aβ bound to HDLs is indeed non-amyloidogenic remains unknown. Furthermore, this study did not address the potential interaction of HDLs with Aβ oligomers or indeed fibrils.

These findings begin to rationalise the subpopulation-specific neuroprotective action of apoA-I-containing HDLs [9] by confirming the presence of a preferred mechanism of interaction of Aβ with apoA-I in HDLs involving strong hydrophobic interactions with the ‘LN’ region as well as an additional network of hydrogen bonds. These structural determinants would specifically mediate Aβ sequestration by apoA-I in HDLs. The therapeutic implications of this interaction relate to enhancing and promoting Aβ clearance mechanisms in AD, reducing or preventing Aβ aggregation and toxicity, as well as potentially identifying peptide mimics of the apoA-I interacting region with Aβ [75].

## Figures and Tables

**Figure 1 ijms-26-01380-f001:**
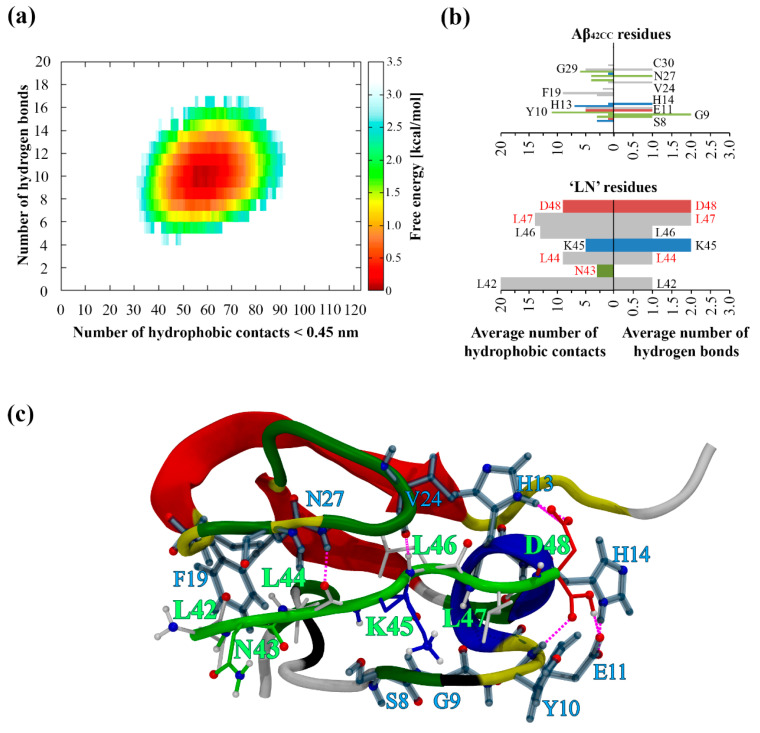
(**a**) Free energy landscape corresponding to the MD simulation of Aβ_42CC_ and ‘LN’ as a function of intermolecular hydrophobic contacts (<0.45 nm) and hydrogen bonds. (**b**) Average number of residue-wise intermolecular hydrophobic contacts (left) and intermolecular hydrogen bonds (right) between Aβ_42CC_ (**top**) and ‘LN’ (**bottom**) throughout the final 150 ns of a conventional MD simulation. Select residues are labelled according to their single-letter amino acid code followed by their residue number. Red lettering corresponds to the residues within ‘LN’ region of apoA-I that are equivalent to the high-affinity Aβ-binding ‘GNLLTLD’ peptide. Horizontal bars are coloured according to general residue typing (basic—blue; acidic—red; non-polar—grey; polar—green). (**c**) Final snapshot of the Aβ_42CC_–‘LN’ complex corresponding to the free energy minimum for the simulation. Mainchain atoms drawn in a cartoon representation with ‘LN’ coloured green and the Aβ_42CC_ peptide coloured according to secondary structure. ‘LN’ sidechains and interacting Aβ_42CC_ residues are drawn in a licorice representation, with ‘LN’ residues coloured according to residue type whilst Aβ_42CC_ residues are all coloured grey. Intermolecular hydrogen bonds are drawn as dashed magenta lines.

**Figure 2 ijms-26-01380-f002:**
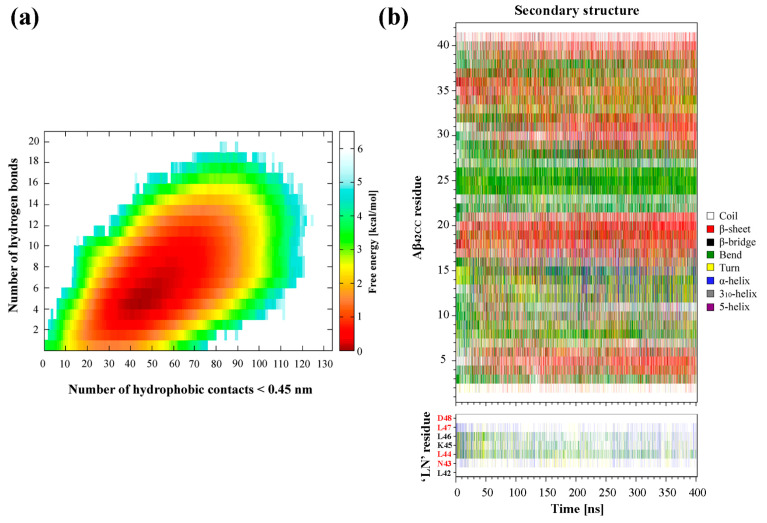
(**a**) Free energy landscape as a function of the number of intermolecular hydrophobic contacts and hydrogen bonds formed between the ‘LN’ fragment and the Aβ_42CC_ peptide at 310 K. (**b**) Secondary structures sampled by the Aβ_42CC_ peptide (**top**) and the ‘LN’ fragment (**bottom**) at 310 K. Red lettering corresponds to the residues within ‘LN’ region of apoA-I that are equivalent to the high-affinity Aβ-binding ‘GNLLTLD’ peptide.

**Figure 3 ijms-26-01380-f003:**
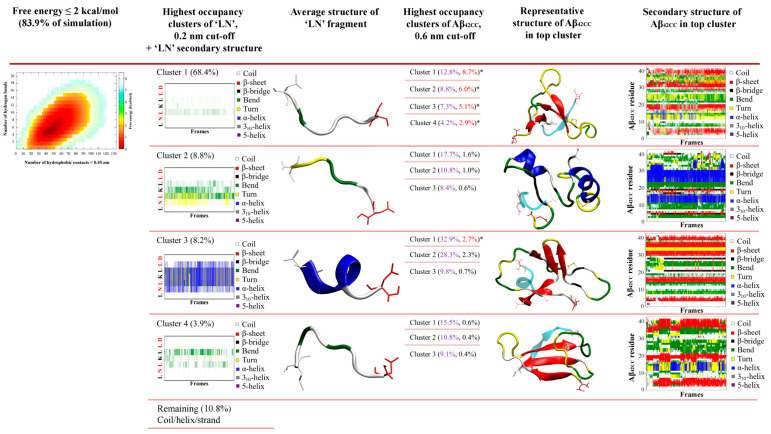
Identification of binding poses between the ‘LN’ fragment of apoA-I and an Aβ_42CC_ peptide at 310 K followed a two-step RMSD clustering regime considering low free energy states (ΔG ≤ 2 kcal/mol, 83.9% of the total replica time) taken from a T-REMD simulation. Initial clustering of ‘LN’ identified four high-occupancy clusters ranging from 3.9 to 68.4% of all low free energy states, followed by a secondary clustering of each ‘LN’ cluster based upon Aβ_42CC_. Bracketed percentages in black and red indicate the overall cluster occupancy as a percentage of the total considered low free energy states, with purple numbering denoting the relative proportion of the given ‘LN’ cluster and red numbering highlighting the top five highest-occupancy poses. Top five poses are denoted by an asterisk. Four of the top five poses were identified within the first ‘LN’ cluster, Cluster 1, wherein ‘LN’ exhibits a predominantly extended coil structure as shown in the corresponding secondary-structure plot. The fifth-highest-occupancy pose came from the mostly helical ‘LN’ Cluster 3. Note: Red lettering in ‘LN’ secondary structure plots correspond to the residues within ‘LN’ region of apoA-I that are equivalent to the high-affinity Aβ-binding ‘GNLLTLD’ peptide.

**Figure 5 ijms-26-01380-f005:**
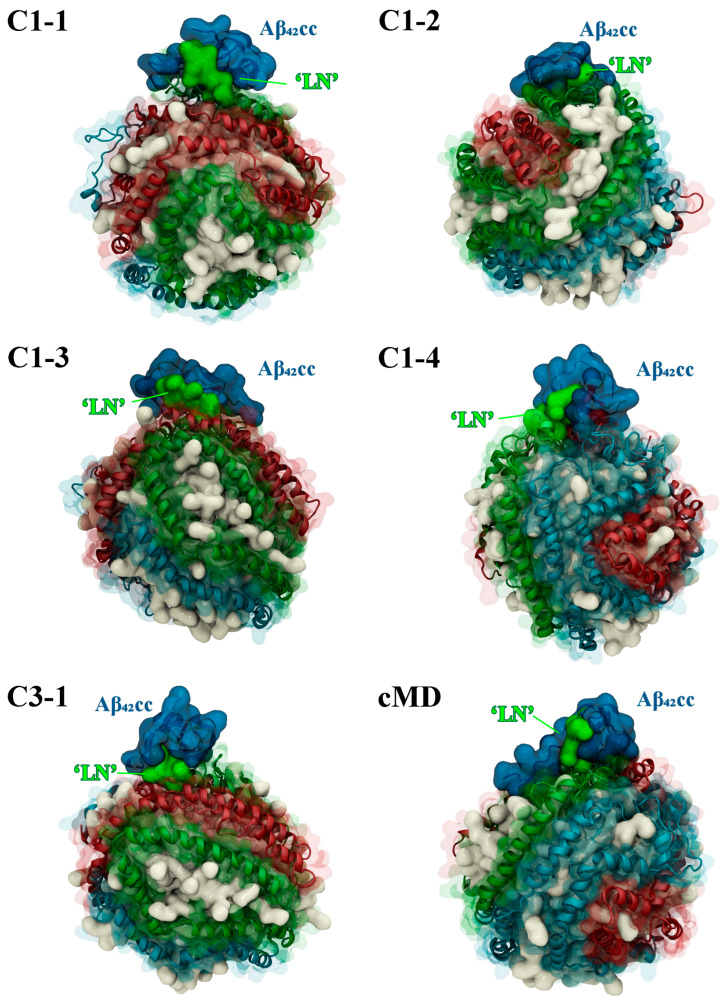
Snapshots of the final MD simulation configuration for each Aβ42CC-HDL system showing that the two species are retained as a complex. Aβ42CC peptides and interacting ‘LN’ segments of the apoA-I chain are drawn in a deep blue and bright green surface, respectively. HDL surface lipids have been drawn as a white opaque surface, with HDL apoA-I chains drawn in an opaque cartoon representation surrounded by a transparent surface representation and each chain has been coloured separately as sky blue, red or green.

**Figure 6 ijms-26-01380-f006:**
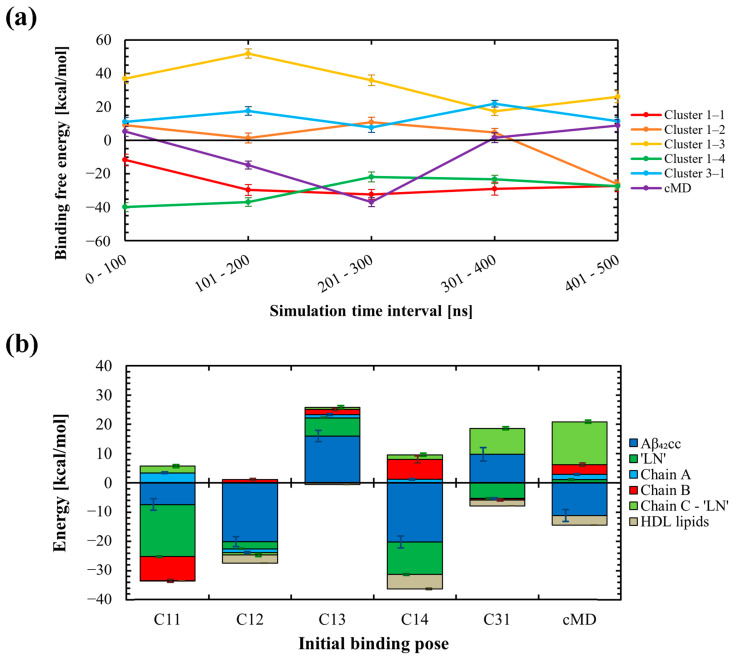
(**a**) Summary of the 100 ns time-interval-wise binding free energies calculated throughout MD simulations of six HDL_3c_-Aβ_42CC_ systems, each starting with a different initial binding pose between Aβ_42CC_ and the full HDL_3c_ particle via the exposed ‘LN’ region of apoA-I. Three initial binding poses (Cluster 1-1, Cluster 1-2, and Cluster 1-4) ultimately exhibited favourable and remarkably similar binding free energy values following 500 ns of MD simulation. The remaining three initial binding poses (Cluster 1-3, Cluster 3-1 and the cMD pose) ultimately exhibited unfavourable binding free energies following 500 ns of conventional MD simulation. (**b**) Decomposition of binding free energies for system components across the six binding poses throughout the final 100 ns of the MD simulations, with relative standard error shown as off-set error bars between the respective stacked bars.

**Figure 7 ijms-26-01380-f007:**
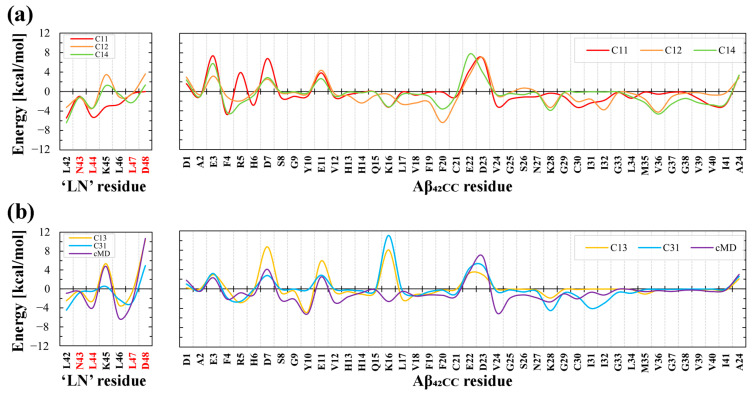
Average residue-wise decomposition of the free energies of binding across the interacting ‘LN’ region of apoA-I and the Aβ_42CC_ peptide during the final 100 ns interval of a 500 ns MD simulation of HDL–Aβ_42CC_ complexes for (**a**) the three favourable binding poses, and (**b**) the three unfavourable binding poses. Red lettering in the labels for ‘LN’ residues correspond to the residues within ‘LN’ region of apoA-I that are equivalent to the high-affinity Aβ-binding ‘GNLLTLD’ peptide.

**Figure 8 ijms-26-01380-f008:**
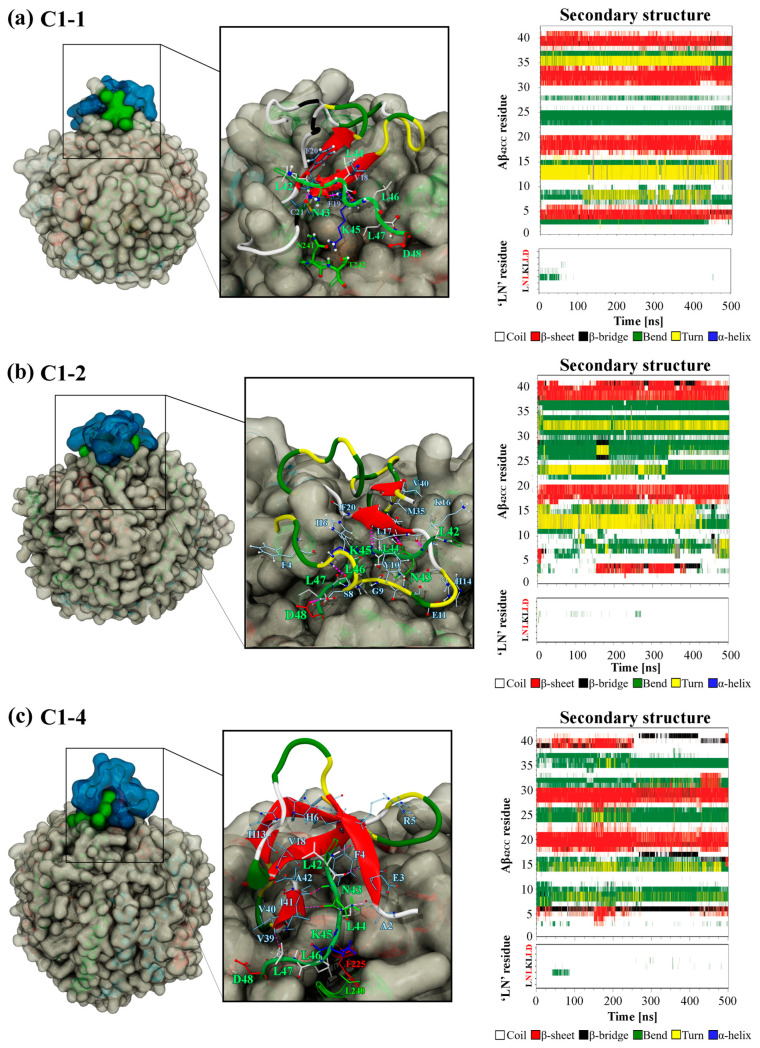
Snapshot of the final MD simulation configuration of the HDL–Aβ_42CC_ complex for (**a**) C1-1, (**b**) C1-2 and (**c**) C1-4, with the surfaces of interacting ‘LN’ and Aβ_42CC_ coloured green and blue, respectively. Insets show the binding interface with interacting residues drawn in a liquorice representation and the protein mainchain drawn in a cartoon representation. Aβ_42CC_ is coloured according to secondary structure motif as defined in corresponding time-wise secondary structure profiles for Aβ_42CC_ as well as the interacting ‘LN’ segment (far right). Note: Red lettering in ‘LN’ secondary structure plots correspond to the residues within ‘LN’ region of apoA-I that are equivalent to the high-affinity Aβ-binding ‘GNLLTLD’ peptide.

**Figure 9 ijms-26-01380-f009:**
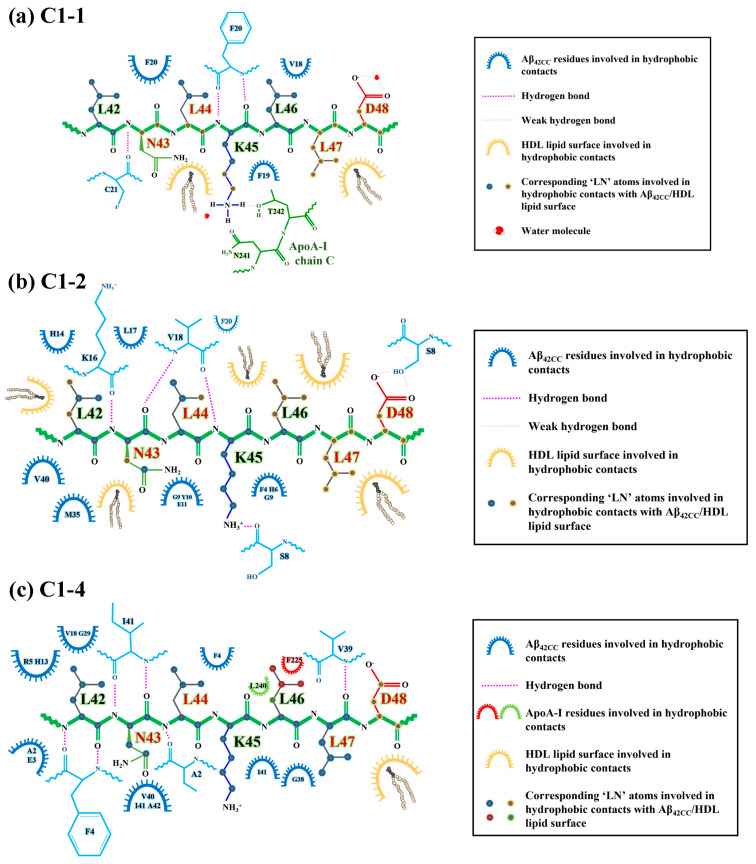
Diagrammatic representation of the intermolecular interactions observed at the end of the HDL simulations in the binding interface of ‘LN’ (green) with Aβ_42CC_ (blue) for (**a**) the (**a**) C1-1, (**b**) C1-2 and (**c**) C1-4 binding poses. Binding interface schematics generated according to data derived from Visual Molecular Dynamics (VMD) v. 1.9.3.

**Figure 10 ijms-26-01380-f010:**
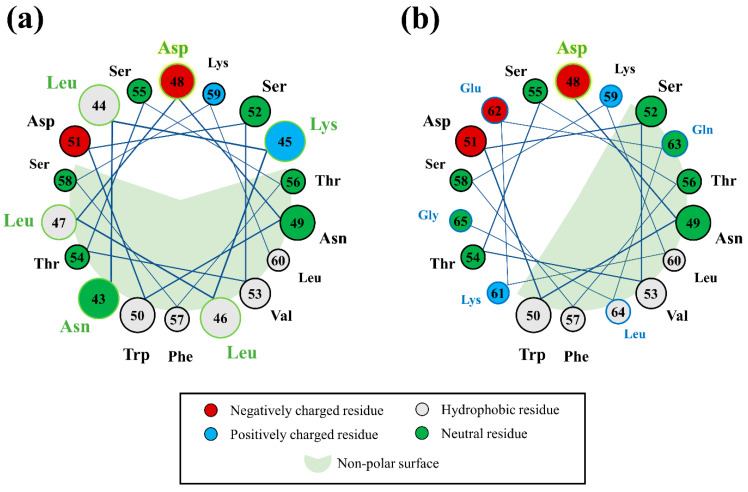
Helical wheel diagrams of the H1 domain of lipidated apoA-I showing the position of Asp48 with respect to the prevailing non-polar face of the helix when residues within the ‘LN’ segment (circles with a green outline and green text) are (**a**) in a helical arrangement, and (**b**) when no longer contributing a part of the helix. Note: The succeeding ^61^KEQLG^65^ segment of H1 has been incorporated into the second helical wheel for context (circles with a blue outline and blue text).

## Data Availability

Data are available on request from the authors.

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
