# Peer review of "Molecular Simulation of the Binding of Amyloid Beta to Apolipoprotein A-I in High-Density Lipoproteins"

_ijms, 2025, doi:10.3390/ijms26031380_

Round 1

Reviewer 1 Report

Comments and Suggestions for Authors

The manuscript by Malajczuk  and Mancera entitled "Molecular simulation of the binding of amyloid beta to 2 apolipoprotein A-I in high density lipoproteins" is interesting and, generally, well-written. While it is well-written, here and there the text due to its AI text enhancement gets a bit "fuzzy", somebody needs to read and fix the manuscript.

Comments on the Quality of English Language

see above

Author Response

Comment 1: The manuscript by Malajczuk  and Mancera entitled "Molecular simulation of the binding of amyloid beta to 2 apolipoprotein A-I in high density lipoproteins" is interesting and, generally, well-written. While it is well-written, here and there the text due to its AI text enhancement gets a bit "fuzzy", somebody needs to read and fix the manuscript.

Response: We thank the reviewer for their approval of our manuscript. We can assure that no AI text enhancement was used at all. Revisions made to the text in light of comments made by other reviewers will hopefully clarify and streamline the text.

Reviewer 2 Report

Comments and Suggestions for Authors

This manuscript explores the interactions between the LN fragment of apolipoprotein A-I (apoA-I) and the Aβ42CC peptide through molecular dynamics (MD) which could be potential therapeutic implications for Alzheimer’s disease (AD). Through a combination of conventional MD and temperature replica exchange MD (T-REMD) simulations, the study investigates the structural determinants that drive the binding of Aβ to high-density lipoproteins (HDLs) mediated by the LN segment which could be stabilized by hdrophobic interactions and hydrogen bonding formaion. Although the study is lack of chemical/biological/biochemcial experimental results, by characterizing multiple conformational states and binding energies of the potent detailed HDL–Aβ interactions, the authors provide potential for the development of therapeutic strategy to treat AD. The writing is straightforward and the experimental/theoretical approaches of the presented research make it attractive to the readership of the International Journal of Molecular Science; however, major revision of the manuscript is necessary prior to publication.

Comments:

1.       Please clearly mention the hypotheses and objectives of this study in the introduction.

2.       The authors applied LN region for the study, however, the entire protein (ApoA) could be different from that segments. It should be discussed further. Also, the authors should consider how LN region affects lipidation or protein heterogeneity in physiological conditions.

3.       It is required to discuss how the identified binding poses could inform the design of therapeutic molecules against AD by clearing Abeta or inhibiting Abeta aggregation. In addition, what could happen in the presence of metal ions (such as Zn ion, Cu ion, Fe ion).

4.       Please provide more detailed discussion of previously reported HDL-Abeta interactions and the new findings from the simulations in this study. Moreover, it would be better for suggesting the experiments (or techniques) for validating the simulations results.

5.       For me, the abstract looks a bit out of focus. Please simplify and emphasize the significance of this study.

6.       In discussion, provide more detailed information of interactions, such as key amino acids, binding sites. Also, at the same time, please delete the repeated contents.

7.       Provide more detail on the force field choices and their suitability for modeling both HDL and Aβ42CC clearly. The importance of the 16KLVFF20 segment of Aβ42CC in binding is discussed but not elaborated upon in the context of other known binding mechanisms of apoA-I. Providing a comparative analysis could strengthen the argument.

8.       Define technical terms such as “16KLVFF20 segment” and “β-hairpin motif” upon their first mention to improve accessibility.

9.       Ensure consistent use of formatting styles, such as italics for gene/protein names and consistent citation styles. Also, put the figure captions in a same direction as their figures.

10.    The conclusion could better contextualize the findings in the broader scope of AD research. Discussing how the identified binding modes could inform therapeutic development would be impactful. In addition, please provide the recommendations for future research.

11.    Please double check the information of references.

Author Response

COMMENT 1: Please clearly mention the hypotheses and objectives of this study in the introduction.

RESPONSE 1: We have modified the last paragraph of the Introduction to make it clearer that our hypothesis is that the binding of Abeta to HDL particles proceeds through hydrophobic interactions with the 'LN' segment of apoA-I, and that that the computational study has the objective of identifying likely interaction modes.

COMMENT 2: The authors applied LN region for the study, however, the entire protein (ApoA) could be different from that segments. It should be discussed further. Also, the authors should consider how LN region affects lipidation or protein heterogeneity in physiological conditions.

RESPONSE 2: The simulation strategy adopted in this work included a 'grafting' stage, whereby five representative binding poses of the Abeta-LN interaction (predicted by T-REMD simulations) were grafted into full HDL3c model particles, where indeed the entire structure of ApoA-I was present (see reference 24). This HDL3c particle conformation was chosen because it exhibits the largest amount of solvent exposure of the 'LN' fragment (as described in reference 24). Upon grafting, subsequent molecular dynamics simulations of the complex of Abeta with the full HDL particle (with the entire structure of apoA-I) for each binding mode were conducted for 1 microsecond. Consequently, the full impact of the entire structure of apoA-I and all lipids present in the HDL3c particle model were taken into account in the characterisation of the binding interactions with Abeta. Our earlier work (references 22-24) on HDL particles fully described the role of lipidation and particle size (affecting the number of apoA-I proteins present) in physiological conditions.

COMMENT 3: It is required to discuss how the identified binding poses could inform the design of therapeutic molecules against AD by clearing Abeta or inhibiting Abeta aggregation. In addition, what could happen in the presence of metal ions (such as Zn ion, Cu ion, Fe ion).

RESPONSE 3: We have expanded the last paragraph of the Conclusions and provided an additional reference to a review paper to emphasise that the main significance of this work relates to confirming the the role of HDLs in Abeta clearance as a therapeutic approach to AD, as well as identifying peptide mimics of the apoA-I interacting region with Abeta to prevent deleterious Abeta aggregation. We are not aware of any specific study of the role of metal ions on the interaction between HDLs and Abeta.

COMMENT 4: Please provide more detailed discussion of previously reported HDL-Abeta interactions and the new findings from the simulations in this study. Moreover, it would be better for suggesting the experiments (or techniques) for validating the simulations results.

RESPONSE 4: To our knowledge, this is the first computational study of the interaction of the HDL-Abeta interaction, culminating our previous work on modelling HDL particle subpopulations (references 22-24). Both the Introduction (e.g. paragraphs 2 and 3) and the Results/Discussion section include discussion of experimental evidence that either supports the notion of the Abeta-'LN' interaction, or that is consistent with the conformations and/or interactions observed in both Abeta and apoA-I ('LN') in support of our predictions. The last paragraph of the Conclusions section provides a summary of the key predictions of this work that would be amenable to further experimental characterisation, such as the predominant conformation of 'LN' when solvent exposed, the changes in conformation of the KLVFF region in Abeta, the lipid interactions with the Asp48 residue of 'LN', and the HDL interactions with the KLVFF region in Abeta. Importantly, these findings are aimed to a provide a mechanistic rationale for experimental observations relating to the known role of HDL in reducing Abeta oligomerisation and the formation of toxic oligomers.

COMMENT 5: For me, the abstract looks a bit out of focus. Please simplify and emphasize the significance of this study.

RESPONSE 5: We have removed parts of the text in the abstract that were indeed unnecessary, and added a final sentence related to the significance of this study, all within the required 200-word limit.

COMMENT 6: In discussion, provide more detailed information of interactions, such as key amino acids, binding sites. Also, at the same time, please delete the repeated contents.

RESPONSE 6: Figure 4 (binding poses predicted by the T-REMD simulation) provide a detailed picture of the time-average interactions of both the 'LN' fragment and Abeta. Unlike in a docking simulation study, the data reflects the average over the simulations at 310K (as opposed to identifying specific binding sites), reflecting the dynamic nature of the underlying interactions. Figure 8 and 9 provide detailed information of the amino acids involved and the type of interactions on both the HDL particle and Abeta for the final configuration in the simulations. It is again important to consider that there is no expectation of identifying a unique, well-defined binding mode (or indeed a "binding site") given the dynamic nature of all molecules involved. Having said, we have suggested in our paper that the 16KLVFF20 region in Abeta in a beta-structure conformation may constitute a specific binding site for the 'LN' segment of apoA-I.

COMMENT 7: Provide more detail on the force field choices and their suitability for modeling both HDL and Aβ42CC clearly. The importance of the 16KLVFF20 segment of Aβ42CC in binding is discussed but not elaborated upon in the context of other known binding mechanisms of apoA-I. Providing a comparative analysis could strengthen the argument.

RESPONSE 7: The second paragraph of section 3.3 provides an explanation for our choice of the GROMOS54A7 force field, both in terms of its ability to improve upon the ability of its predecessor (GROMOS53A6) to reproduce beta-hairpin conformations in Abeta and better represent protein helical propensities, but also to sample and stabilise the folding of beta-peptides. Furthermore, the choice of force field parameters for lipids is consistent with our previous work on the development of full HDL particles. We note that the use of Abeta42CC was aimed at facilitating the formation of a stable beta-hairpin conformation, which was indeed observed throughout the simulations, as explained in section 3.1. The focus on the 16KLVFF20 segment in Abeta arises not only because of its central role in amyloidogenesis, but also because it was predicted to play a critical role in both hydrophobic and hydrogen-bonding interactions with the 'LN' fragment. There does not appear to be any other specific information about regions in Abeta that mediate the interaction with apoA-I.

COMMENT 8: Define technical terms such as “16KLVFF20 segment” and “β-hairpin motif” upon their first mention to improve accessibility.

RESPONSE 8: We have modified the manuscript to explain what a beta-hairpin motif is and to emphasise that the 16KLVFF20 segment is the amyloidogenic core region of Abeta.

COMMENT 9: Ensure consistent use of formatting styles, such as italics for gene/protein names and consistent citation styles. Also, put the figure captions in a same direction as their figures.

RESPONSE 9: We have checked that the citation style is consistent, and that our formatting style is in accordance with MDPI formatting. As regards the location of the figure captions, the MDPI editorial team will decide where best to place them.

COMMENT 10: The conclusion could better contextualize the findings in the broader scope of AD research. Discussing how the identified binding modes could inform therapeutic development would be impactful. In addition, please provide the recommendations for future research.

RESPONSE 10: We have added a new final paragraph in the Conclusions section to address this point in succinct form,

COMMENT 11: Please double check the information of references.

RESPONSE 11: All references have been double checked.

Reviewer 3 Report

Comments and Suggestions for Authors

The authors studied six atomistic models as possible representations of the complex between an engineered amyloid-beta peptide (Abeta42CC), mimicking a beta-rich population of toxic Abeta42, and the HDL3c particle, where the apoA-I protein exposes the 'LN' loop.
The Abeta42 peptide is locked by the disulfide bond present in the engineered sequence: A21C/A30C mutant.
As for the HDL particle the authors used the smallest one obtained in the work of Ref.24 (HDL3c).
They made a careful construction of 'LN' loops using the highest populated conformers obtained combining the 300 ns long MD simulation of Abeta42CC/LN complex (sect. 2.1) and 400 ns-long T-REMD simulation (sect. 2.2).

The results of the modelling, though affected by the, mostly unavoidable, statistical limitations, are of great interest to researchers involved in understanding the network of interactions of amyloid-beta peptides.
I recommend a series of changes, mainly about the length of the manuscript.

The manuscript is very long and, because of the reported (and largely expected) statistical limitations, it can be made shorter to focus on the relevant results reported in sect. 2.3.
Sections 2.1-2.2 describe the preparation, with all reported limitations, of the poses and they can be moved to supplementary information.

Section 2.1 -
As stated by authors (lines 211-215), conclusions of sect. 2.1 (lines 184-197) are largely biased by the single encounter complex that was modelled in the MD simulation.
The complex formation is rapid and the model is clearly a single representation of the complex collapse.
This result was expected.
The reader at this point does not understand why the authors did not make an attempt to simulate the Abeta42CC/LN complex formation by T-REMD simulations (sect. 2.2) since the beginning.

Section 2.2 -
By T-REMD simulations the authors discovered that the affinity between the two peptides is so large that dissociation never occurs even at high temperature.
This is a common issue in T-REMD.
To partially overtake this bias the authors should sparse the initial configurations of the 48 replica since the beginning.
Possibly using more than 48 replica.

Section 2.3 -
In the assessment of thermodynamic stability of the six chosen binding poses the authors should remind that entropy differences between solute configurations are neglected, see Eq. 1 in one of the first MM-PBSA publications:

https://doi.org/10.1021/ar000033j

The authors should mention that computational methods able to associate/dissociate ligands forming complexes are indeed available.
The authors should state since the beginning that their approach is the only affordable one because of the large size of the involved HDL particle (even though it is the smallest oobserved in Ref.24).

Lines 469-482 well describe the fact that the work is a first approximate assessment of structural determinants of stability or instability.

A comparison between the Abeta42CC configuration used in the manuscript and reference structures reported in the literature for Abeta42 monomers might be of great interest to readers.
There are reference monomers: those present in the fibril structures as well as those stabilizing soluble Abeta42 oligomers (dimers, tetramers, dodecamers).
This is particularly important because of the relevance of electrostatic interactions that is found in the manuscript, for instance between 'LN' and chain C of ApoA-I as competing with Abeta42CC/LN.

As for Conclusions, the finding of determinants (lines 859-866) is the major one.
What follows, about the importance of 16-20 region in Abeta42 in amyloid aggregation, is misleading.
This because the segment is critical for the formation of fibrils, but its role in the formation of toxic (not necessarily amyloidogenic) oligomers is debated.
Other regions are important in toxicity, because stabilising soluble potentially toxic oligomers.
Therefore, lines 866-874 should change, because the stabilisation of non-amyloidogenic conformations can eventually enhance toxicity.

Author Response

COMMENT 1: The manuscript is very long and, because of the reported (and largely expected) statistical limitations, it can be made shorter to focus on the relevant results reported in sect. 2.3.
Sections 2.1-2.2 describe the preparation, with all reported limitations, of the poses and they can be moved to supplementary information.

RESPONSE 1: We accept that the manuscript is somewhat long, but for a computational study we think it is important to retain section 2.1, which reports the results of a conventional, unbiased MD simulation of the interaction of Abeta with the 'LN' fragment. Some of the results are indeed reported in the Supporting Material already. In any case, section 2.1 is not that long. As regards section 2.2., we believe that it must stay in the paper because it reports the details of the T-REMD simulations, from which the key five binding poses reported in Figure 3 are derived. Consequently we do not believe that sections 2.1-2.2 should be removed.

COMMENT 2: Section 2.1 - As stated by authors (lines 211-215), conclusions of sect. 2.1 (lines 184-197) are largely biased by the single encounter complex that was modelled in the MD simulation. The complex formation is rapid and the model is clearly a single representation of the complex collapse. This result was expected. The reader at this point does not understand why the authors did not make an attempt to simulate the Abeta42CC/LN complex formation by T-REMD simulations (sect. 2.2) since the beginning.

RESPONSE 2: We thank the reviewer for this opinion; however, we did not in fact expect the rapid formation of a complex between the two species, nor, more interestingly, that it would remain stable throughout the length of the simulation. Both molecules were initially placed well apart from each other, at a distance of 20 Angstroms (2.0 nm), so it was interesting to observe the relatively rapid interaction between the two species. There are multiple computational studies in the literature based exclusively on conventional MD simulations of the interaction of disordered proteins which, in our opinion (and no doubt that of the reviewer), lack robustness in the approach followed. Consequently, it is interesting to provide here comparative evidence of the role of enhanced sampling (section 2.3).

COMMENT 3: Section 2.2 - By T-REMD simulations the authors discovered that the affinity between the two peptides is so large that dissociation never occurs even at high temperature. This is a common issue in T-REMD. To partially overtake this bias the authors should sparse the initial configurations of the 48 replica since the beginning. Possibly using more than 48 replica.

RESPONSE 3: The choice in the number of replicas and temperature range was done according to the algorithm of Patriksson and van der Spoel, as explained in the paper, which led to an average exchange probability between replicas of 22% (compared to the expected 20%), whilst ensuring that the T-REMD simulation had converged. While it is possible that a different number of replicas might have resulted in the lack of complex formation (or dissociation) at the highest temperatures, that is unlikely to result in major changes in the formation of binding complexes at the more favourable lower temperatures. All of our subsequent clustering analysis and selection of binding poses was carried out at 310K. Inevitably, repeating the T-REMD simulations would lead to repeating this entire study, which is not practically feasible.

COMMENT 4: Section 2.3 - In the assessment of thermodynamic stability of the six chosen binding poses the authors should remind that entropy differences between solute configurations are neglected, see Eq. 1 in one of the first MM-PBSA publications: https://doi.org/10.1021/ar000033j

RESPONSE 4: The reviewer is absolutely right to point this out. The standard MM-PBSA method does not include any estimate of entropy (which, in this case, is primarily the vibrational entropy) in the formation of binding complexes. We have added a statement to this effect at the end of the Methods section.

COMMENT 5: The authors should mention that computational methods able to associate/dissociate ligands forming complexes are indeed available. The authors should state since the beginning that their approach is the only affordable one because of the large size of the involved HDL particle (even though it is the smallest observed in Ref.24).

RESPONSE 5: We agree with this comment. We have added a statement to this effect at the end of the Methods section.

COMMENT 6: Lines 469-482 well describe the fact that the work is a first approximate assessment of structural determinants of stability or instability.

RESPONSE 6: Thank you. We indeed agree with this assessment of the limitations that we have stated about our own work.

COMMENT 7: A comparison between the Abeta42CC configuration used in the manuscript and reference structures reported in the literature for Abeta42 monomers might be of great interest to readers. There are reference monomers: those present in the fibril structures as well as those stabilizing soluble Abeta42 oligomers (dimers, tetramers, dodecamers). This is particularly important because of the relevance of electrostatic interactions that is found in the manuscript, for instance between 'LN' and chain C of ApoA-I as competing with Abeta42CC/LN.

RESPONSE 7: We thank the reviewer for this interesting comment. Most experimental structures relate to solid state/crystal structures, rather than to disordered conformations that can be observed during the early stages of aggregation. We have indeed published a comprehensive study of the conformations of monomeric Abeta42 in solution (reference 30), but did not make a direct comparison with this work. The second part of section 2.2 discusses in some detail the conformational diversity of Abeta42CC observed in the five representative binding modes arising from the dominant conformational ensembles. Because of the competing interaction of the 'LN' region with chain C of apoA-I, as the reviewer points out, the mechanism (i.e. conformations) of interaction of Abeta42 with apoA-I may not be related to the self-aggregation of Abeta42.

COMMENT 8: As for Conclusions, the finding of determinants (lines 859-866) is the major one. What follows, about the importance of 16-20 region in Abeta42 in amyloid aggregation, is misleading. This because the segment is critical for the formation of fibrils, but its role in the formation of toxic (not necessarily amyloidogenic) oligomers is debated. Other regions are important in toxicity, because stabilising soluble potentially toxic oligomers. Therefore, lines 866-874 should change, because the stabilisation of non-amyloidogenic conformations can eventually enhance toxicity.

RESPONSE 8: We agree with the reviewer that this is contested territory. We have modified the relevant paragraph in the Conclusions to note this uncertainty. In addition, we have acknowledged that this study did not investigate the potential interaction of HDLs with Abeta oligomers or fibrils.

Reviewer 4 Report

Comments and Suggestions for Authors

This is an interesting simulation study performed with high resolution techniques aimied at investigating the relationship among beta-amyoid and  apoA-I-containing HDLs.

the study is well written and very precise, but I suggest authors to increase the translationality of their findings. in order ot increase its appeal also in clinical readers.

- There have been several studies which investigated in clinical populations of AD patinetsthe relationship between APoE polimorphism and different cytokynes profiles individuating specific links also in terms of clinical progressions (DOI:10.3233/JAD-191250). DO authors think that a possible invovlment of inflammation derivates might alter (or influence) the interaction between amyloid and ApoA-Icontaining HDLs.?

Author Response

Comment 1: 

This is an interesting simulation study performed with high resolution techniques aimied at investigating the relationship among beta-amyoid and  apoA-I-containing HDLs. The study is well written and very precise, but I suggest authors to increase the translationality of their findings. in order ot increase its appeal also in clinical readers.

- There have been several studies which investigated in clinical populations of AD patinets the relationship between APoE polimorphism and different cytokynes profiles individuating specific links also in terms of clinical progressions (DOI:10.3233/JAD-191250). DO authors think that a possible invovlment of inflammation derivates might alter (or influence) the interaction between amyloid and ApoA-Icontaining HDLs?

Response: We completely agree with the reviewer's comment about the importance of ApoE polymorphism in AD clinical progression, which is well established in the literature. Our work, however, focuses on a separate aspect: the early stage of the disease where the role of lipid/cholesterol metabolism can play a critical role in AD pathogenesis. In this case, our work relates to the potentially protective role of Abeta binding to HDL particles via ApoA-I. It is thus not possible to establish if there is a potential link between inflammatory molecules and this specific interaction without additional experimental information, which is outside of the scope of this work.

Round 2

Reviewer 2 Report

Comments and Suggestions for Authors

This manuscript has been revised based on my previous comments. However, still some points should be revised.

1. Figure 3 and 4 looks like tables. Label them as table. Also, the text in "secondary structure" column should be larger. It is difficult to read. (So, it would be better to make new figures or tables for them.)

2. Figure 8, the right side images should have larger font for the text.

Author Response

Comments 1: Figure 3 and 4 looks like tables. Label them as table. Also, the text in "secondary structure" column should be larger. It is difficult to read. (So, it would be better to make new figures or tables for them.)

Response 1: We appreciate your feedback on Figures 3 and 4. We have increased the font size in the “secondary structure” column to improve readability, as you suggested. However, we respectfully maintain that Figures 3 and 4 are more appropriately labeled as figures rather than tables. These are composite images designed to visually convey information in a logical and intuitive manner, and reformatting them as tables may detract from their clarity and impact. We hope the improvements made to the font size address the concerns regarding readability.

Comment 2: Figure 8, the right side images should have larger font for the text.

Response 2: Thank you for highlighting the need for improved readability in Figure 8. We have increased the font size for the text on the right-side images to ensure better clarity and accessibility. We believe these adjustments meet these requirements.

We are grateful for your thoughtful suggestions, which have helped us enhance the overall presentation of our work.

Reviewer 3 Report

Comments and Suggestions for Authors

The authors did not answer in the manuscript text most of my questions.
Therefore, I recommend a further minor revision.

Response 1: I leave to Editors any decision among the opportunity to keep Section 2.1 and to make the manuscript shorter (and clearer).

Responses 2-3: The authors based their work on the hypothesis that the binding of Abeta42CC to HDL particles proceeds through hydrophobic interactions (lines 141-147 in version 2, lines 138-142 in version 1).
In dynamic models, like conventional MD, separate hydrophobic patches in water rapidly begin to interact the one with each other.
T-REMD methods, designed to overtake this limitation, usually sparse initial configurations in order to reduce the bias.
Notice that this is additional to sparsing temperature values.
According to Methods (lines 723-736, version 2) we understand that the 48 replica share the same coordinates and have only different initial velocities (temperature).
This setting does not reduce the bias.
What I recommended was just to warn the reader of this limitation for future studies.
Of course, I did not recommend to repeat the whole T-REMD simulation.

Responses 4-6: The authors added text to comment about.

Responses 7-8: Since the authors have a wide population of monomeric Abeta42 (ref. 30), they can easily discuss/summarize which is the selection operated by the three steps:
i) Abeta42CC modification of Abeta42;
ii) Abeta42CC/LN interactions;
iii) Abeta42CC/HDL interface.
This discussion would help the clarity of the manuscript as for the Abeta42/HDL aggregation-competing mechanism.

Author Response

Comments 1: I leave to Editors any decision among the opportunity to keep Section 2.1 and to make the manuscript shorter (and clearer).

Response 1: We appreciate the reviewer’s feedback regarding the length and clarity of Section 2.1. To reiterate our earlier response, we believe that this section provides essential context and background information, which is important for understanding the subsequent simulations and analysis.

Comments 2: The authors based their work on the hypothesis that the binding of Abeta42CC to HDL particles proceeds through hydrophobic interactions (lines 141-147 in version 2, lines 138-142 in version 1).
In dynamic models, like conventional MD, separate hydrophobic patches in water rapidly begin to interact the one with each other.
T-REMD methods, designed to overtake this limitation, usually sparse initial configurations in order to reduce the bias.
Notice that this is additional to sparsing temperature values.
According to Methods (lines 723-736, version 2) we understand that the 48 replica share the same coordinates and have only different initial velocities (temperature).
This setting does not reduce the bias.
What I recommended was just to warn the reader of this limitation for future studies.
Of course, I did not recommend to repeat the whole T-REMD simulation.

Response 2: We acknowledge the Reviewer’s point that initialising the 48 replicas with the same coordinates but different velocities does not fully address potential biases related to hydrophobic interactions, as separate hydrophobic patches may rapidly interact in water during conventional MD. We also appreciate the clarification that the intention was not to recommend repeating the entire T-REMD simulation but rather to highlight this as a limitation for future studies.

While starting all replicas from the same Aβ42CC conformation could, in principle, limit initial conformational diversity, the extensive simulation time (19.2 µs) and frequent temperature exchanges in our T-REMD simulations facilitated broad conformational sampling as evidenced by the resulting free energy landscape and conformational diversity presented in Figure 2. We have now explained in the revised manuscript that this suggests that potential biases were not a significant limitation in our study.

To further address this, we have expanded the relevant Results and Discussion section (lines 289-295 in the revised manuscript). Specifically, we now emphasise that the broad free energy landscape (Figure 2a) and extensive secondary structure diversity (Figure 2b) demonstrate that the T-REMD regime was effective in overcoming any potential biases associated with initialising all replicas from the same structural conformation. This reinforces the validity of our approach while maintaining transparency in our methodological considerations. Furthermore, we also acknowledge that future studies may consider initialising replicas from multiple distinct conformations to further enhance sampling efficiency, and have noted this as a potential avenue for improvement in the revised manuscript.

Comment 3: Since the authors have a wide population of monomeric Abeta42 (ref. 30), they can easily discuss/summarize which is the selection operated by the three steps:
i) Abeta42CC modification of Abeta42;
ii) Abeta42CC/LN interactions;
iii) Abeta42CC/HDL interface.
This discussion would help the clarity of the manuscript as for the Abeta42/HDL aggregation-competing mechanism.

Response 3: We appreciate the reviewer’s suggestion to delineate the respective Aβ42CC conformations contributing to (i) its β-hairpin structure, (ii) its interaction with the ‘LN’ peptide, and (iii) its interaction with HDL particles. While this is an interesting avenue of inquiry, we believe that such an analysis is beyond the scope of the present study.

The engineered β-hairpin structure of Aβ42CC, which has been argued to be a biologically relevant model for neurotoxic Aβ (Refs. 14-16), imposes conformational constraints that limit the extent of structural transitions observed during these interactions. As such, a systematic comparison of different Aβ42CC conformations across the sequential interaction stages was not a central aim of this study.

Furthermore, while Sonar et al. (Ref. 30) investigated the wild-type Aβ42 conformational landscape, their system lacked the 'LN' peptide of ApoA-I/a full HDL, each of which strongly influence the conformational sampling of Aβ42CC in our simulations. As a result, and in combination with the differences between the engineered Aβ42CC and wild-type Aβ42, a direct comparison between the structures observed in their study and those present in our simulations would be methodologically problematic.

We have ensured that our discussion appropriately acknowledges both the biological relevance and limitations of Aβ42CC as a model system, as well as the role of hydrophobic exposure in driving aggregation, citing Sonar et al. where relevant. However, we believe that a detailed dissection of Aβ42CC conformations at different interaction stages would require a dedicated study and is outside the scope of the present work.

Round 3

Reviewer 2 Report

Comments and Suggestions for Authors

This manuscript has been revised by the authors based on the comments from the reviewer.

This revised manuscript looks now acceptable for publication.